# Deterministic genetic barcoding for multiplexed behavioral and single-cell transcriptomic studies

**Jorge Blanco Mendana[1], Margaret Donovan[1], Lindsey Gengelbach O'Brien[1], Benjamin Auch[1], John Garbe[1], Daryl M Gohl[1,2]\***

[1]University of Minnesota Genomics Center, Minneapolis, Minneapolis, United States; [2]Department of Genetics, Cell Biology, and Development, University of Minnesota, Minneapolis, United States

**eLife Assessment**

This **useful** study presents a genetically encoded barcoding system that could advance transcriptomic studies and that has the potential for further applications, such as in high-throughput population-scale behavioral measurements. The evidence supporting the claims of the authors is **solid** and highlights both the usefulness and the limitations of the approach.

**\*For correspondence:**
dmgohl@umn.edu

**Competing interest:** The authors declare that no competing interests exist.

**Abstract** Advances in single-cell sequencing technologies have provided novel insights into the dynamics of gene expression and cellular heterogeneity within tissues and have enabled the construction of transcriptomic cell atlases. However, linking anatomical information to transcriptomic data and positively identifying the cell types that correspond to gene expression clusters in single-cell sequencing data sets remains a challenge. We describe a straightforward genetic barcoding approach that takes advantage of the powerful genetic tools in *Drosophila* to allow in vivo tagging of defined cell populations. This method, called T̲argeted G̲enetically-E̲ncoded M̲ultiplexing (TaG-EM), involves inserting a DNA barcode just upstream of the polyadenylation site in a Gal4-inducible *UAS-GFP* construct so that the barcode sequence can be read out during single-cell sequencing, labeling a cell population of interest. By creating many such independently barcoded fly strains, TaG-EM enables positive identification of cell types in cell atlas projects, identification of multiplet droplets, and barcoding of experimental timepoints, conditions, and replicates. Furthermore, we demonstrate that TaG-EM barcodes can be read out using next-generation sequencing to facilitate population-scale behavioral measurements. Thus, TaG-EM has the potential to enable large-scale behavioral screens in addition to improving the ability to multiplex and reliably annotate single-cell transcriptomic experiments.

## Introduction

Spatially and temporally regulated gene expression patterns are a hallmark of multicellular life and function to orchestrate patterning, growth, and differentiation throughout development (*Ingham, 1988*; *Reeves et al., 2006*). In mature organisms, spatial expression patterns both in tissues and within cells define functionally distinct compartments and determine many aspects of cellular and organismal physiology (*Martin and Ephrussi, 2009*). In addition, such expression patterns differentiate healthy and diseased tissue and impact disease etiology (*Marusyk et al., 2012*). Spatial and temporal expression patterns, which can be used to distinguish between cell types and provide insight into cellular function, also provide a means to understand the organization and physiology of complex tissues

**eLife digest** From delivery to shipping or shopping, barcodes are a part of everyday life. In biological research as well, 'barcoding' cells and organisms using specific DNA sequences has been a transformative approach. Such tags can be introduced into the genetic material of cells, allowing scientists to label cell populations or individuals of interest.

Here, Mendana et al. investigated how DNA barcoding could be used to cut down the time and cost required to pinpoint a certain population of cells, or of organisms, within a larger group. At present, such efforts often remain labor intensive and costly. For instance, it is now possible for researchers to capture all the genes that are switched on at any given time in individual cells in an organism; however, it is still difficult to then identify which tissue or population of interest a particular cell belongs to.

In response, Mendana et al. established a new approach in fruit flies, called TaG-EM, which makes it possible to bypass these limitations by introducing a carefully designed genetic barcode, easily read by DNA sequencers, into the genome of the fly. Further experiments also demonstrated that TaG-EM was valuable at the scale of an organism, to be used in behavioral experiments. Typically, researchers examine how various strains of animals respond to different conditions by testing each group separately; Mendana et al. were able to show that 'barcoding' the flies using TaG-EM made it possible to pool these behavioral measurements, as the different groups could then be later quickly identified using their genetic tags. Overall, this new approach should allow researchers using fruit flies to investigate questions around gene expression and behavior in a faster and cheaper way, improving our understanding of a range of biological processes.

such as the brain (*Thompson et al., 2014*). Thus, robust and scalable tools for measuring spatial and temporal gene expression patterns at a genome-wide scale and at high resolution would be transformative research tools across many biological disciplines.

Single-cell sequencing technologies have provided insights into the dynamics of gene expression throughout development, been used to characterize somatic variation and heterogeneity within tissues, and are currently enabling the construction of transcriptomic cell atlases (*Klein et al., 2015*; *Macosko et al., 2015*; *Zheng et al., 2017*). However, linking anatomical information to transcriptomic data and positively identifying the cell types that correspond to gene expression clusters in single-cell sequencing data sets remains a challenge. The cellular identities of gene expression clusters identified in cell atlas data sets are typically inferred from the expression of distinctive gene sets (*Hung et al., 2020*; *Li et al., 2022*; *Ma et al., 2021*), and the lack of positive identification of gene expression clusters introduces an element of uncertainty in this analysis. Moreover, this process of manual annotation is labor-intensive and often requires additional experiments to determine or confirm the expression patterns of marker genes. Emerging spatial genomics technologies hold promise in linking anatomical and transcriptomic information (*Lee et al., 2015*; *Lein et al., 2017*). Several of the emerging commercial spatial genomics technologies rely on in situ sequencing of marker genes allowing droplet-based single-cell transcriptomic data to be mapped onto a tissue. However, these technologies currently suffer from constraints related to cost, content, or applicability to specific model systems.

In addition to descriptive cell atlas projects, studies involving multiple experimental timepoints throughout development and aging, or studies assessing the effects of experimental exposures or genetic manipulations would benefit from increased ability to multiplex samples. Given the fixed costs of droplet-based single-cell sequencing, generating data for single-cell transcriptomic time courses or experimental manipulations can be costly. Outside of descriptive studies, these costs are also a barrier to including replicates to assess biological variability; consequently, a lack of biological replicates derived from independent samples is a common shortcoming of single-cell sequencing experiments. Antibody-based cell hashing or feature barcoding approaches have been developed to allow multiplexing of samples in droplet-based single-cell sequencing reactions (*Stoeckius et al., 2018*; *Stoeckius et al., 2017*). In addition, other multiplexing strategies for single-cell sequencing based on alternative methods for tagging cells (*Cheng et al., 2021*) or making use of natural genetic variation have been used (*Kurmangaliyev et al., 2020*). While such approaches can reduce per-sample costs,

typically samples are barcoded at a population level and thus do not enable labeling of cell subpopulations within a sample.

We have developed a straightforward genetic barcoding approach that takes advantage of the powerful genetic tools available in *Drosophila* to allow deterministic in vivo tagging of defined cell populations. This method, called *T*argeted *G*enetically-*E*ncoded *M*ultiplexing (TaG-EM), involves inserting a DNA barcode just upstream of the poly-adenylation site in a Gal4-inducible *UAS-GFP* construct so that the barcode sequence can be read out during droplet-based single-cell sequencing, labeling a cell population of interest.

Genetic barcoding approaches have been employed in many unicellular systems, cell culture, and viral transfection to facilitate high-throughput screening using sequencing-based readouts (*Bhang et al., 2015*; *Smith et al., 2009*; *van Opijnen et al., 2009*). In multicellular animals, techniques such as GESTALT have enabled lineage tracing by using CRISPR to create unique barcodes in differentiating tissue (*McKenna et al., 2016*), and barcode sequencing has also been employed to map connectivity in the brain (*Chen et al., 2019*).

Genetically barcoded fly lines can also be used to enable highly multiplexed behavioral assays which can be read out using high-throughput sequencing. Flies carrying TaG-EM barcodes can be exposed to different experimental perturbations and then tested in assays where flies, larvae, or embryos are fractionated based on behavioral outcomes or other phenotypes. Thus, TaG-EM has the potential to enable large-scale next-generation sequencing (NGS)-based behavioral or other fractionation screens analogous to BAR-Seq or Tn-Seq approaches employed in microbial organisms.

## Results

### TaG-EM: A novel genetic barcoding strategy for multiplexed behavioral and single-cell transcriptomics

We cloned a fragment containing a PCR handle sequence and a diverse 14 bp barcode sequence into the SV40 3' untranslated region (UTR) sequence just upstream of the polyadenylation sites in the 10xUAS-myr::GFP (pJFRC12, *Pfeiffer et al., 2010*) backbone (*Figure 1A*). A pool containing 29 unique barcode-containing plasmids was injected into *Drosophila* embryos for PhiC31-mediated integration into the attP2 landing site (*Groth et al., 2004*) and transgenic lines were isolated and confirmed by Sanger sequencing (*Figure 1B*, *Figure 1—figure supplement 1*). We recovered 20 distinctly barcoded *Drosophila* lines, with some barcodes recovered from multiple crosses (*Figure 1—figure supplement 1*). Such barcoded fly lines have the potential to enable population behavioral measurements, where different exposures, experimental timepoints, and genetic or neural perturbations can be multiplexed and analyzed by measuring barcode abundance in sequencing data (*Figure 1C*). In addition, the barcodes, which reside on a Gal4-inducible *UAS-GFP* construct, can be expressed tissue-specifically and read out during droplet-based single-cell sequencing, labeling a cell population and/or an experimental condition of interest (*Figure 1D*).

### Testing the accuracy and reproducibility of TaG-EM behavioral measurements using structured pools

We conducted initial experiments to optimize amplification of the genetic barcodes using primers targeting the PCR handle inserted just upstream of the 14 bp barcode sequence and PCR primers downstream of the TaG-EM barcode in the SV40 3' UTR sequence (*Figure 2—figure supplement 1*). To test the accuracy and reproducibility of sequencing-based measurements of TaG-EM barcodes, we constructed structured pools containing defined numbers of flies pooled either evenly with each of the 20 barcode constructs comprising 5% of the pool, or in a staggered manner with sets of barcodes differing in abundance in 2-fold increments (*Figure 2A*). To examine the impact of technical steps such as DNA extraction and PCR amplification on TaG-EM barcode measurements, even pools were made and extracted in triplicate and amplicon sequencing libraries were made in triplicate for each independently extracted DNA sample for both the even and staggered pools. The resulting data indicated that TaG-EM measurements are highly accurate and reproducible. Technical replicates (indicated by error bars in *Figure 2B–E*) showed minimal variability. Likewise, the three independently extracted replicates of the even pools produced consistent data with all 20 barcodes detected at levels close to the expected 5% abundance (*Figure 2B–C*). Barcode abundance values for the staggered structured

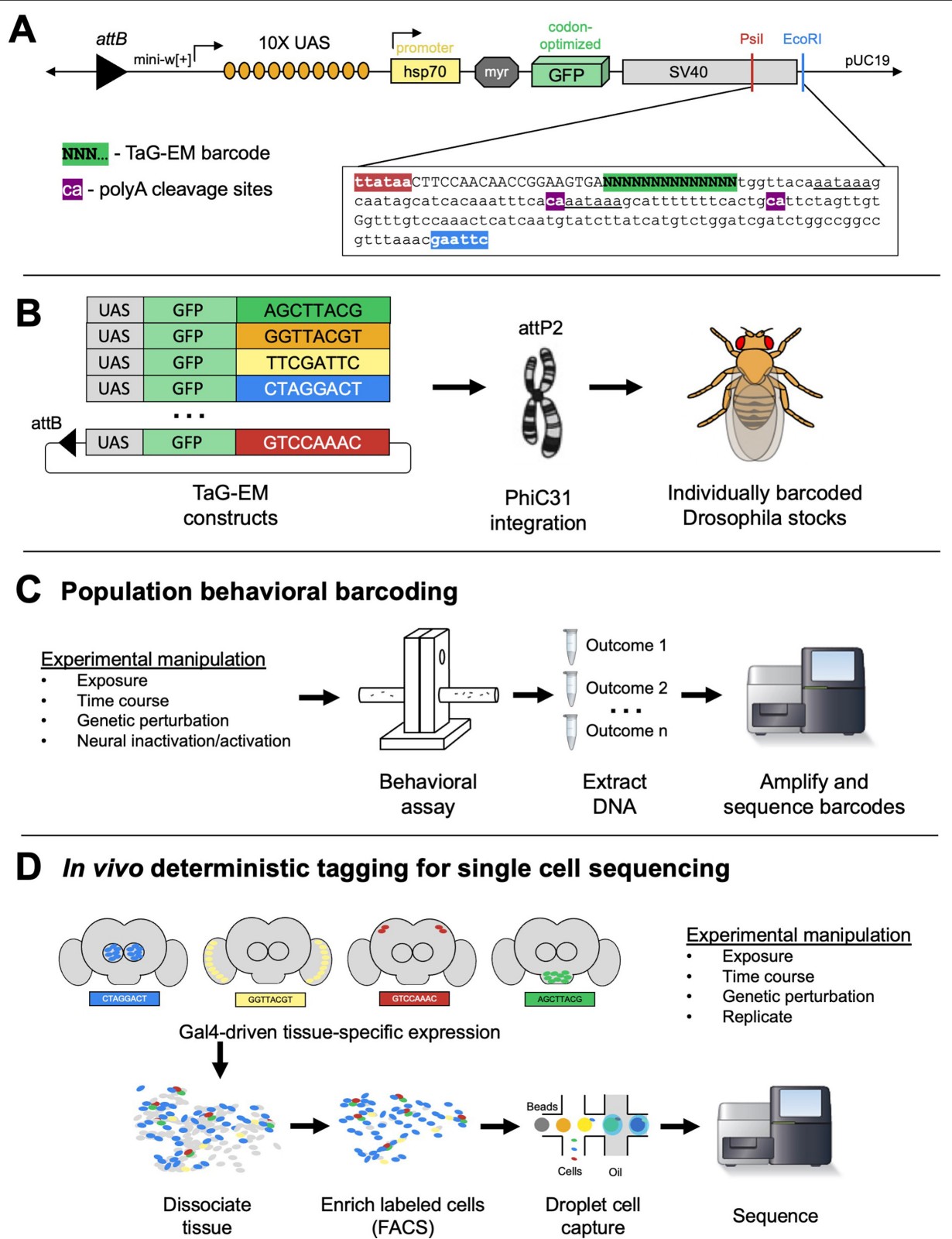

**Figure 1.** Overview of TaG-EM system. (**A**) Detailed view of the 3′ UTR of the TaG-EM constructs showing the position of the 14 bp barcode sequence (green highlight) relative to the polyadenylation signal sequences (underlined) and poly-A cleavage sites (purple highlights). The pJFRC12 backbone schematic is modified with permission from an unpublished schematic made by Barret Pfeiffer. (**B**) Schematic illustrating the design of the TaG-EM constructs, where a barcode sequence is inserted in the 3′ UTR of a UAS-GFP construct and inserted in a specific genomic locus using PhiC31 integrase.

*Figure 1 continued on next page*

*Figure 1 continued*

(**C**) Use of TaG-EM barcodes for sequencing-based population behavioral assays. (**D**) Use of TaG-EM barcodes expressed with tissue-specific Gal4 drivers to label cell populations in vivo upstream of cell isolation and single-cell sequencing.

The online version of this article includes the following figure supplement(s) for figure 1:

**Figure supplement 1.** Sanger sequencing identification of TaG-EM barcode lines.

pools was generally consistent with the input values and in most cases, the twofold differences between the different groups of barcodes could be distinguished (*Figure 2D–E*). The coefficients of variation were largely consistent for groups of TaG-EM barcodes pooled evenly or at different levels within the staggered pools (*Figure 2—figure supplement 2*). For the staggered pools, abundances correlated well with the expected values, particularly when multiple barcodes for an input level were averaged, in which case $R^2$ values were >0.99 (*Figure 2D–E*, inset plots). This indicates that a high level of quantitative accuracy can be attained using sequencing-based analysis of TaG-EM barcode abundance, particularly when averaging data for three to four independent barcodes for an experimental condition.

## TaG-EM measurement of phototaxis behavior correlate well with video-based measurements

Next, we tested whether TaG-EM could be used to measure a phototaxis behavior. A mixture of barcoded wild type or blind *norpA* mutant flies were run together through a phototaxis assay. At the end of a period of light exposure, test tubes facing toward or away from the light were capped, DNA was isolated, and barcodes were amplified and sequenced for each tube. Raw read counts were scaled in proportion to the number of flies per tube and a preference index was calculated for each barcode (*Figure 3A*). In parallel, individual preference indices were calculated based on manual scoring of videos recorded for each line (*Figure 3B*). Preference indices calculated for the pooled, NGS-based TaG-EM measurements were nearly identical to conventional behavioral measurements for both wild type and *norpA* mutants (*Figure 3A–B*).

## TaG-EM measurement of oviposition behavior and age-dependent fecundity

We next tested whether NGS-based pooled measurements of egg laying could be made. Fertilized females from each of the 20 barcode lines were placed together in egg laying cups, embryos were collected, aged for 12 hr to enable cell numbers to stabilize in the developing eggs, and then DNA was extracted from both the pooled adult flies and the embryos. In general, TaG-EM measurements of oviposition correlated with fly numbers, with the exception of barcode 14 which had reduced barcode abundance across multiple trials (*Figure 3—figure supplement 1*). This suggests that despite the fact that the genetic barcode constructs are inserted in a common landing site, differences with respect to specific behaviors may exist among the lines, and thus one should test to make sure given lines are appropriate to use in specific behavioral assays.

To determine whether TaG-EM could be used to measure age-dependent fecundity, we collected flies from twelve different TaG-EM barcode lines at four time points separated by 1 week (three barcode lines per timepoint). We collected eggs from these fly lines individually and scored the number of viable eggs per female. Next, we pooled the barcoded flies from all timepoints and collected eggs from the pooled flies. These eggs were aged, DNA was extracted, and the TaG-EM barcodes were amplified and sequenced. While measurements from individual barcode lines were noisy, both for manual counts and sequencing based measurements, there was a general trend toward declining fecundity over time (*Figure 3—figure supplement 2*), consistent with published reports (*David et al., 1975*). Manually scored viable egg numbers and TaG-EM barcode abundances were well correlated across two independent experimental trials ($R^2$ values of 0.52–0.61 for Trial 1 and 0.74–0.84 for Trial 2). When barcodes from each individual timepoint were averaged, $R^2$ values for the correlation between manual and sequencing-based measurements were 0.95 for Trial 1 and 0.99 for Trial 2 (*Figure 3C*, *Figure 3—figure supplement 3*).

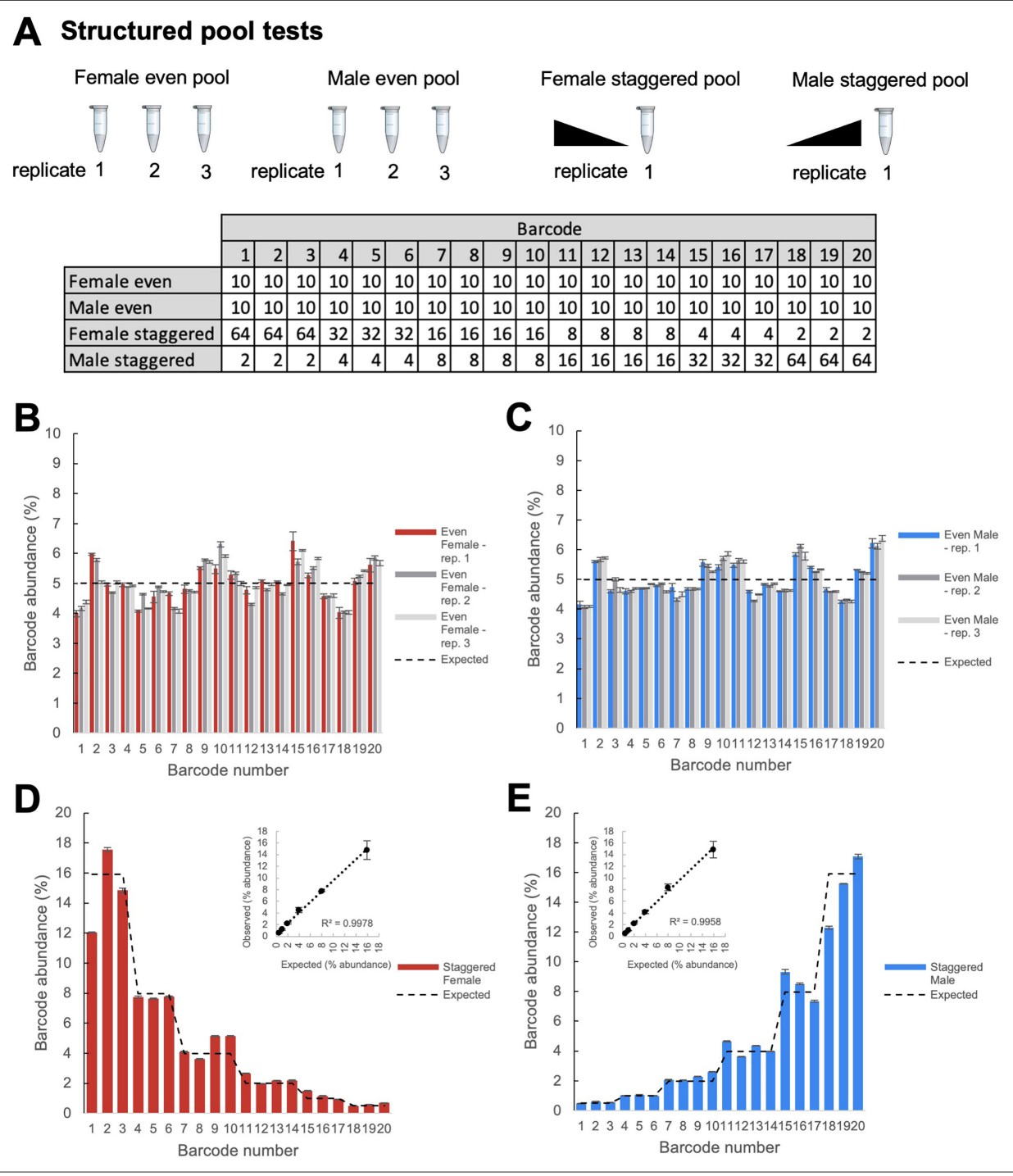

**Figure 2.** Structured pool tests. (**A**) Overview of the construction of the structured pools for assessing the quantitative accuracy of TaG-EM barcode measurements. Male and female even pools were constructed and extracted in triplicate. The table shows the number of flies that were pooled for each experimental condition. (**B**) Barcode abundance data for three independent replicates of the female even pool. (**C**) Barcode abundance data for three independent replicates of the male even pool. (**D**) Barcode abundance data for the female staggered pool. Inset plot shows the average observed barcode abundance among lines pooled at each level compared to the expected abundance. (**E**) Barcode abundance data for the male staggered pool. Inset plot shows the average observed barcode abundance among lines pooled at each level compared to the expected abundance. For all plots, bars indicate the mean barcode abundance for three technical replicates of each pool, error bars are +/-S.E.M.

The online version of this article includes the following source data and figure supplement(s) for figure 2:

**Figure supplement 1.** Optimization of TaG-EM barcode amplification.

*Figure 2 continued on next page*

*Figure 2 continued*

**Figure supplement 1—source data 1.** Uncropped gel images with labels for data displayed in *Figure 2—figure supplement 1A*.

**Figure supplement 1—source data 2.** Original files for gel images displayed in *Figure 2—figure supplement 1A*.

**Figure supplement 2.** Coefficient of variation for TaG-EM structured pools.

## Quantifying food transit time in the larval gut using TaG-EM

Gut motility defects underlie a number of functional gastrointestinal disorders in humans (*Keller et al., 2018*). To study gut motility in *Drosophila*, we have developed an assay based on the time it takes a food bolus to transit the larval gut (*Figure 4A*), similar to approaches that have been employed for studying the role of the microbiome in human gut motility (*Asnicar et al., 2021*). Third instar larvae were starved for 90 min and then fed food containing a blue dye. After 60 min, larvae in which a blue bolus of food was visible were transferred to plates containing non-dyed food, and food transit (indicated by loss of the blue food bolus) was scored every 30 min for 5 hr (*Figure 4—figure supplement 1*).

Because this assay is highly labor-intensive and requires hands-on effort for the entire 5-hr observation period, there is a limit on how many conditions or replicates can be scored in one session (~8 plates maximum). Thus, we decided to test whether food transit could be quantified in a more streamlined and scalable fashion by using TaG-EM (*Figure 4B*). Using the manual assay, we observed that while caffeine-containing food is aversive to larvae, the presence of caffeine reduces transit time through the gut (*Figure 4C*, *Figure 4—figure supplement 1*). This is consistent with previous observations in adult flies that bitter compounds (including caffeine) activate enteric neurons via serotonin-mediated signaling and promote gut motility (*Yao and Scott, 2022*). We tested whether TaG-EM could be used to measure the effect of caffeine on food transit time in larvae. As with prior behavioral tests, the TaG-EM data recapitulated the results seen in the manual assay (*Figure 4D*). Conducting the transit assay via TaG-EM enables several labor-saving steps. First, rather than counting the number of larvae with and without a food bolus at each time point, one simply needs to transfer non-bolus-containing larvae to a collection tube. Second, because the TaG-EM lines are genetically barcoded, all the conditions can be tested at once on a single plate, removing the need to separately count each replicate of each experimental condition. This reduces the hands-on time for the assay to just a few minutes per hour. A summary of the anticipated cost and labor savings for the TaG-EM-based food transit assay is shown in *Figure 4—figure supplement 2*.

## Tissue-specific expression of TaG-EM GFP constructs

To facilitate representation of the TaG-EM barcodes in single-cell sequencing data, genetic barcodes were placed just upstream of the polyadenylation signal sequences and poly-A cleavage sites (*Figure 1A*). To verify that the inserted sequences did not interfere with Gal4-driven GFP expression, we crossed each of the barcoded TaG-EM lines to *decapentaplegic-Gal4* (*dpp-Gal4*). We observed GFP expression in the expected characteristic central stripe (*Teleman and Cohen, 2000*) in the wing imaginal disc for 19/20 lines at similar expression levels to the base pJFRC12 *UAS-myr::GFP* construct inserted in the same landing site (*Figure 5A*, *Figure 5—figure supplement 1*). No GFP expression was visible for TaG-EM barcode number 8, which upon molecular characterization had an 853 bp deletion within the GFP coding region (data not shown). We generated and tested GFP expression of an additional 156 TaG-EM barcode lines (*Alegria et al., 2024*), by crossing them to Mhc-Gal4 and observing expression in the adult thorax. All 156 additional TaG-EM lines had robust GFP expression (data not shown). Gal4-driven expression levels of TaG-EM barcoded GFP constructs were also similar to that of the pJFRC12 base construct for multiple driver lines (*Figure 5—figure supplement 2*) indicating that the presence of the barcode does not generally impair expression of GFP.

## Boosting the GFP signal of TaG-EM constructs to enable robust cell sorting

While with some driver lines, expression of the myr::GFP from the TaG-EM construct may be too weak to allow robust enrichment of the tagged cells, adding an additional hexameric GFP construct (*Shearin et al., 2014*) could boost expression of weak driver lines to levels that are sufficient for robust detection of labeled flies or larvae (*Figure 5B*) and for labeling of dissociated cells for flow cytometry

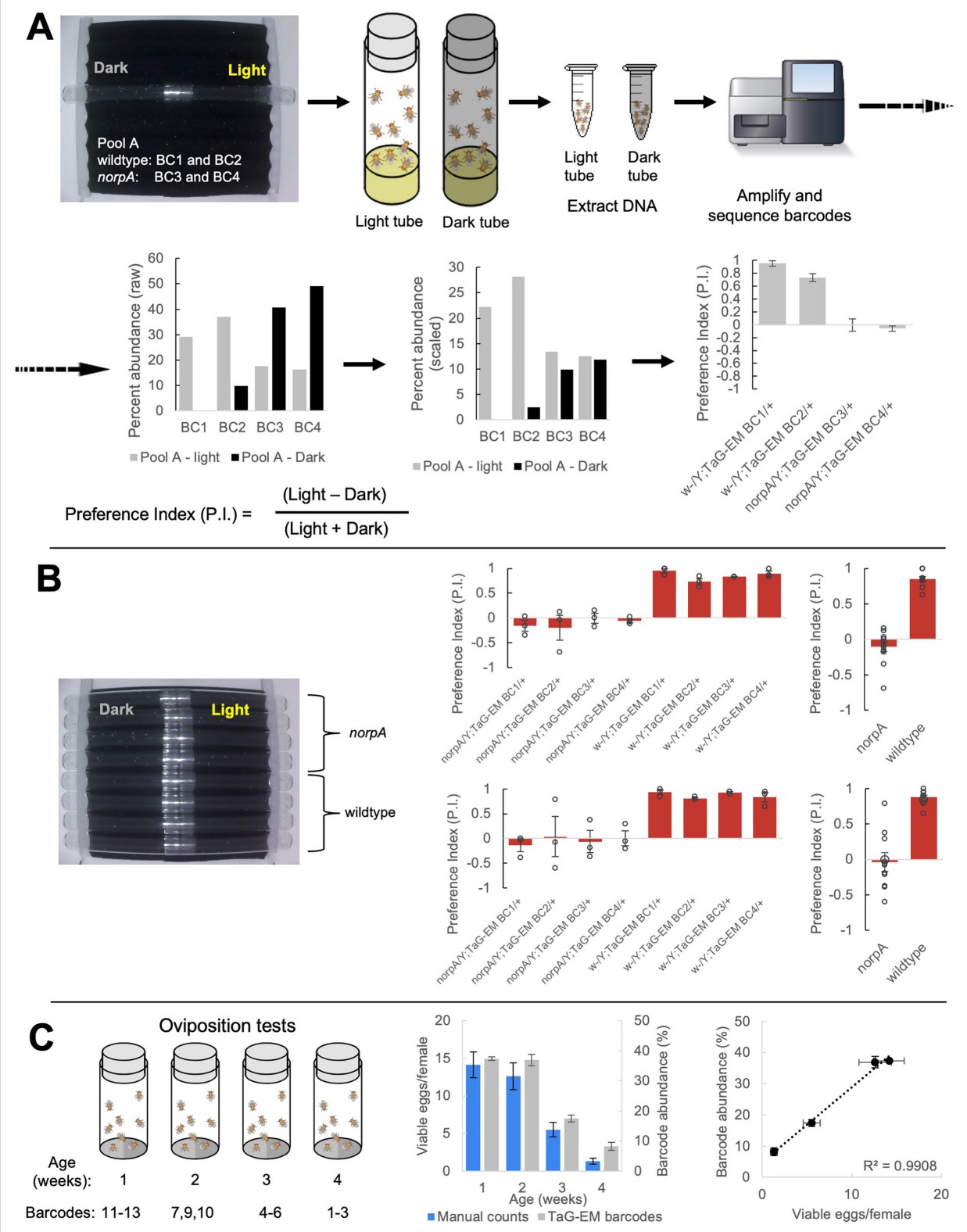

**Figure 3.** TaG-EM barcode-based behavioral measurements. (**A**) TaG-EM barcode lines in either a wild-type or *norpA* background were pooled and tested in a phototaxis assay. After 30 s of light exposure, flies in tubes facing the light or dark side of the chamber were collected, DNA was extracted, and TaG-EM barcodes were amplified and sequenced. Barcode abundance values were scaled to the number of flies in each tube and used to calculate a preference index (P.I.). Average P.I. values for four different TaG-EM barcode lines in both the wild-type and *norpA* backgrounds are shown (n=3 biological replicates, error bars are +/-S.E.M.). (**B**) The same eight lines used for the sequencing-based TaG-EM barcode measurements

*Figure 3 continued on next page*

were independently tested in the phototaxis assay and manually scored videos were used to calculate a P.I. for each genotype. Average P.I. values for each line are shown (n=3 biological replicates, error bars are +/-S.E.M.) for TaG-EM-based quantification (top) and manual video-based quantification (bottom). (C) Flies carrying different TaG-EM barcodes were collected and aged for 1 to 4 weeks and then eggs were collected, and egg number and viability was manually scored for each line. In parallel, the barcoded flies from each timepoint were pooled, and eggs were collected, aged, and DNA was extracted, followed by TaG-EM barcode amplification and sequencing. Average number of viable eggs per female (manual counts) and average barcode abundance are shown both as a bar plot and scatter plot (n=3 biological replicates for 3 barcodes per condition, error bars are +/-S.E.M.).

The online version of this article includes the following figure supplement(s) for figure 3:

**Figure supplement 1.** Oviposition tests with TaG-EM barcode lines.

**Figure supplement 2.** Fecundity data for individual TaG-EM lines.

**Figure supplement 3.** Average age-dependent fecundity data for Trial 1.

(*Figure 6—figure supplement 1*). Stocks with an additional UAS hexameric GFP construct recombined onto the same chromosome as the TaG-EM construct have been established for 20 TaG-EM barcode lines.

## Correlation between expression of TaG-EM barcodes and intestinal cell marker genes in single-cell sequencing data

To test whether we could detect TaG-EM barcodes in single-cell sequencing data, we crossed three TaG-EM barcode lines to two different gut Gal4 driver lines (*Ariyapala et al., 2020*), one expressing in the enterocytes (EC-Gal4: TaG-EM barcodes 1, 2, and 3) and the other in intestinal precursor cells (PC-Gal4: TaG-EM barcodes 7, 8, and 9), which includes stem cells and enteroblasts (EBs). Due to weak GFP expression with the EC-Gal4 driver, we did not see visible GFP positive cells for this driver line. The PC-Gal4 driver line contained an additional UAS-Stinger (2xGFP) construct and expressed GFP at a level sufficient for flow sorting when crossed to the TaG-EM line (*Figure 6—figure supplement 2*). Larval guts were dissected, dissociated, stained with propidium iodide (PI) to label dead cells, and flow sorted to recover PI-negative and GFP-positive cells. Approximately 10,000 cells were loaded into a 10x Genomics droplet generator and a single-cell library was prepared and sequenced. Two clusters were observed in the resulting sequencing data, one of which had high read counts from mitochondrial genes suggesting that this cluster consisted of mitochondria, debris, or dead and dying cells. After filtering the cells with high mitochondrial reads, a single cluster remained (*Figure 6—figure supplement 3*). This cluster expressed known intestinal precursor cell markers such as *escargot* (*esg*), *klumpfuss* (*klu*), and *Notch* pathway genes like *E(spl)mbeta-HLH* (*Figure 6—figure supplement 3*). Expression of all three PC-Gal4-driven TaG-EM barcodes was observed in this cluster (*Figure 6—figure supplement 3*) indicating that TaG-EM barcodes can be detected in single-cell sequencing data. Interestingly, TaG-EM barcode 8, for which no GFP expression was observed, was represented in the single-cell sequencing data indicating that the lesion in the GFP coding region does not prevent mRNA expression for this line.

A previous study used droplet-based single-cell sequencing to characterize the cell types that make up the adult midgut (*Hung et al., 2020*). This study took advantage of two fluorescent protein markers, an *escargot* (*esg*)-GFP fusion protein and a *prospero* (*pros*)-Gal4-driven RFP to label the intestinal stem cells (ISCs) and enteroendocrine cells (EEs), respectively (*Hung et al., 2020*). The authors compared the resulting clusters to a list of known marker genes in the literature, including antibody staining, GFP, LacZ, and Gal4 reporter expression patterns to classify the cells in individual clusters, and also found that the *esg*-GFP expression was present in a broader subset of cells than anticipated. Thus, most of these cell classifications relied upon inference as opposed to direct positive labeling. Recently, a large collection of split-*Gal4* lines were screened for expression in the adult and larval gut (*Ariyapala et al., 2020*). These include pan-midgut driver lines, split-*Gal4* lines specific for the EBs, ECs, EEs, and ISC/EBs, as well as driver lines with regionalized gene expression. We crossed four different TaG-EM barcode lines with the pan-midgut driver (PMG-Gal4: TaG-EM barcodes 1, 2, 3, and 7), and one barcode line to each of the precursor cell (PC-Gal4: TaG-EM barcode 5), enterocyte (EC-Gal4: TaG-EM barcode 4), enteroblast (EB-Gal4: TaG-EM barcode 6), and enteroendocrine (EE-Gal4: TaG-EM barcode 9) drivers (*Ariyapala et al., 2020*). Larval guts were dissected from these lines and cells were dissociated, flow sorted as described above to select live, GFP-positive cells,

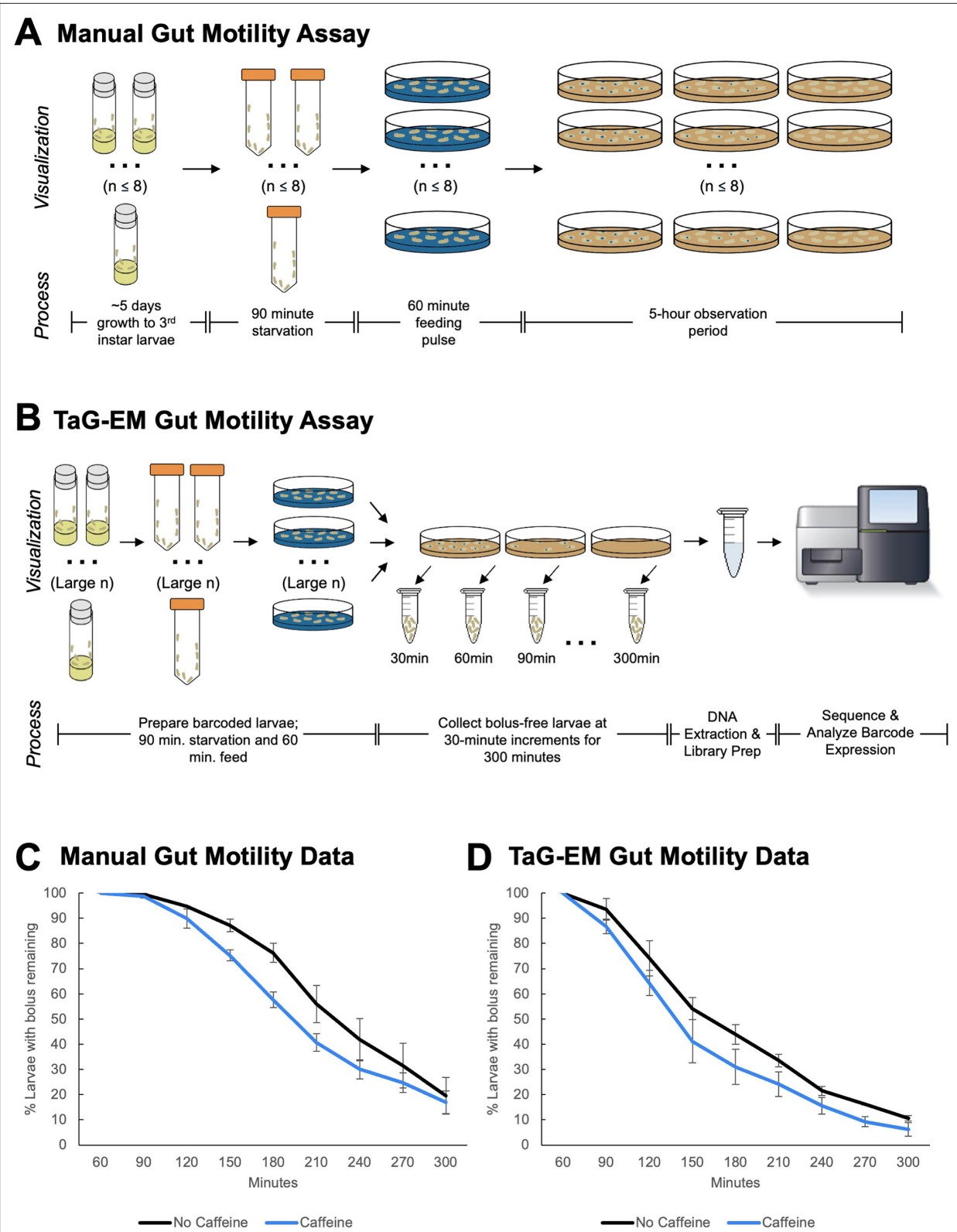

**Figure 4.** TaG-EM barcode-based quantification of larval gut motility. Schematics depicting (**A**) manual and (**B**) TaG-EM-based assays for quantifying food transit time in *Drosophila* larvae. (**C**) Transit time of a food bolus in the presence and absence of caffeine measured using the manual assay (p=0.0340). (**D**) Transit time of a food bolus in the presence and absence of caffeine measured using the TaG-EM assay (p=0.0488). n=3 biological replicates for each condition. A modified Chi-squared method was used for statistical testing (*Hristova and Wimley, 2023*).

*Figure 4 continued on next page*

*Figure 4 continued*
The online version of this article includes the following figure supplement(s) for figure 4:

**Figure supplement 1.** Larval gut motility assay parameters.

**Figure supplement 2.** Cost comparisons for manual and TaG-EM gut motility assays.

and approximately 30,000 cells were loaded into a 10x Genomics droplet generator for single-cell sequencing (*Figure 6—figure supplement 4*). Using the additional hexameric GFP construct to boost GFP expression resulted in visible fluorescent signal for all eight barcode Gal4 line combinations.

An advantage of cell barcoding both for cell hashing (*Stoeckius et al., 2018*) and for TaG-EM in vivo barcoding is that such labeling facilitates the identification and removal of multiplets, which are an artifact of droplet-based single-cell sequencing approaches. After filtering and removing cells with a high percentage of mitochondrial or ribosomal reads, we used DoubletFinder (*McGinnis et al., 2019*) to computationally identify multiplet droplets. In parallel, we searched for cells that co-expressed

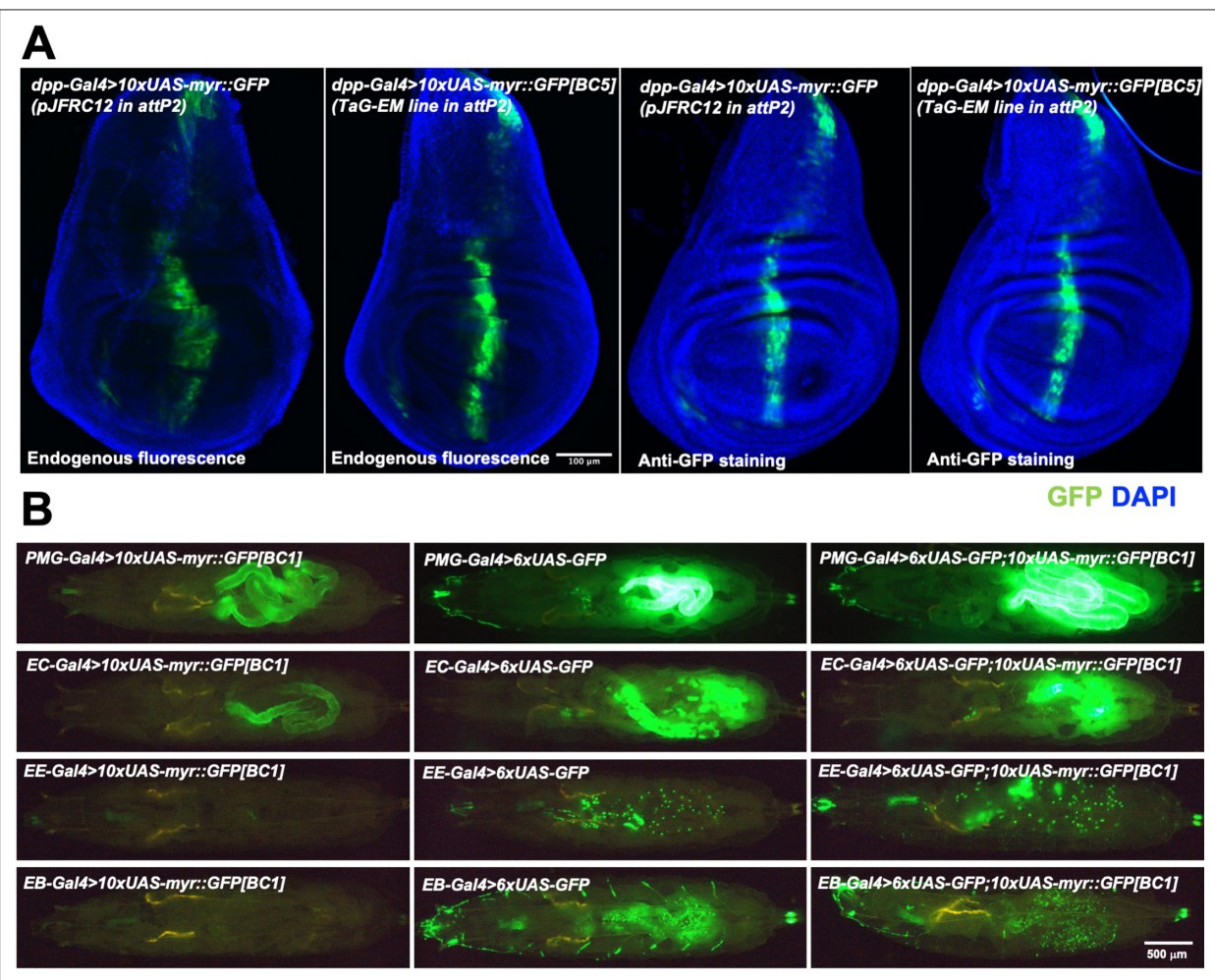

**Figure 5.** Gal4-driven expression of GFP from TaG-EM lines. (**A**) Comparison of endogenous GFP expression and GFP antibody staining in the wing imaginal disc for the original pJFRC12 construct inserted in the attP2 landing site or for a TaG-EM barcode line driven by *dpp-Gal4*. Wing discs are counterstained with DAPI. (**B**) Endogenous expression of GFP from either a TaG-EM barcode construct (left column), a hexameric GFP construct (middle column), or a line carrying both a TaG-EM barcode construct and a hexameric GFP construct (right column) driven by the indicated gut driver line (PMG-Gal4: Pan-midgut driver; EC-Gal4: Enterocyte driver; EE-Gal4: Enteroendocrine driver; EB-Gal4: Enteroblast driver).

The online version of this article includes the following figure supplement(s) for figure 5:

**Figure supplement 1.** Expression driven by *dpp-Gal4* for 20 TaG-EM lines.

**Figure supplement 2.** TaG-EM line GFP expression driven by different Gal4 drivers.

multiple TaG-EM barcodes. DoubletFinder identified 2019 multiplet droplets, while TaG-EM barcodes identified 298 such droplets, 198 of which (66.4%) overlapped with those identified by DoubletFinder (*Figure 6—figure supplement 5*). Thus, TaG-EM help identify an additional 100 doublets that would have otherwise been overlooked using computational doublet identification methods.

After doublet removal, the remaining cells were clustered (*Figure 6A*, *Figure 6—figure supplement 6*) and analyzed using Seurat (*Satija et al., 2015*). Analysis of differentially expressed genes identified clusters expressing marker genes previously reported for adult gut cell types (*Hung et al., 2020*). These included genes associated with precursor cells (*Notch* pathway genes), enterocytes (trypsins, serine proteases, amalyse, mannosidases), and enteroendocrine cells neuropeptides and neuropeptide receptors; (*Figure 6—figure supplement 7*, data not shown). TaG-EM barcodes derived from the eight multiplexed genotypes were observed in approximately one-quarter of the cells (*Figure 6—figure supplement 7*).

In antibody-conjugated oligo cell hashing approaches, sparsity of barcode representation is overcome by spiking in an additional primer at the cDNA amplification step and amplifying the hashtag oligo by PCR. We employed a similar approach to attempt to enrich for TaG-EM barcodes in an additional library sequenced separately from the 10x Genomics gene expression library. Our initial attempts at barcode enrichment using spike-in and enrichment primers corresponding to the TaG-EM PCR handle were unsuccessful (*Figure 6—figure supplement 8*). However, we subsequently optimized the TaG-EM barcode enrichment by (1) using a longer spike-in primer that more closely matches the annealing temperature used during the 10x Genomics cDNA creation step, and (2) using a nested PCR approach to amplify the cell-barcode and unique molecular identifier (UMI)-labeled TaG-EM barcodes (*Figure 6—figure supplement 8*).

Using the enriched library, TaG-EM barcodes were detected in nearly 100% of the cells at high sequencing depths (*Figure 6—figure supplement 9*). However, although we used a polymerase that has been engineered to have high processivity and that has been shown to reduce the formation of chimeric reads in other contexts (*Gohl et al., 2016*), it is possible that PCR chimeras could lead to unreliable detection events for some cells. Indeed, many cells had a mixture of barcodes detected with low counts and single or low numbers of associated UMIs. To assess the reliability of detection, we analyzed the correlation between barcodes detected in the gene expression library and the enriched TaG-EM barcode library as a function of the purity of TaG-EM barcode detection for each cell (the percentage of the most abundant detected TaG-EM barcode, *Figure 6—figure supplement 9*). For TaG-EM barcode detections where the most abundance barcode was a high percentage of the total barcode reads detected (~75%–99.99%), there was a high correlation between the barcode detected in the gene expression library and the enriched TaG-EM barcode library. Below this threshold, the correlation was substantially reduced.

In the enriched library, we identified 26.8% of cells with a TaG-EM barcode reliably detected, a very modest improvement over the gene expression library alone (23.96%), indicating that at least for this experiment, the main constraint is sufficient expression of the TaG-EM barcode and not detection. To identify TaG-EM barcodes in the combined data set, we counted a positive detection as any barcode either identified in the gene expression library or any barcode identified in the enriched library with a purity of >75%. In the case of conflicting barcode calls, we assigned the barcode that was detected directly in the gene expression library. This increased the total fraction of cells where a barcode was identified to approximately 37% (*Figure 6B*).

As expected, the barcodes expressed by the pan-midgut driver were broadly distributed across the cell clusters (*Figure 6—figure supplement 10*). However, the number of cells recovered varied significantly among the four pan-midgut driver associated barcodes. Expression of the cell-type-specific barcodes showed more restricted patterns of expression among the cell clusters and were co-localized with known marker genes for these cell types (*Figure 6C–N*). For instance, TaG-EM barcode 6, driven by the EB-Gal4 line, was expressed primarily in cells that were annotated as enteroblasts (*Figure 6C–D*) and that expressed precursor cell markers such as *esg* (*Figure 6E*), *klu* (*Figure 6F*), and Notch pathway genes such as *e(spl)mbeta-HLH* and *e(spl)m3-HLH* (not shown).

TaG-EM barcode 4 expression, which was driven by the EC-Gal4 line, was seen primarily in a cluster that was annotated as enterocytes (*Figure 6G–H*) and that expressed enterocyte markers such as the serine protease *Jon99Ciii* and other enterocyte marker genes such as the amylase, *Amy-d* (not shown), but not the *beta-Trypsin (betaTry)* gene (*Figure 6I–J*). Detailed characterization of the EC-Gal4 line

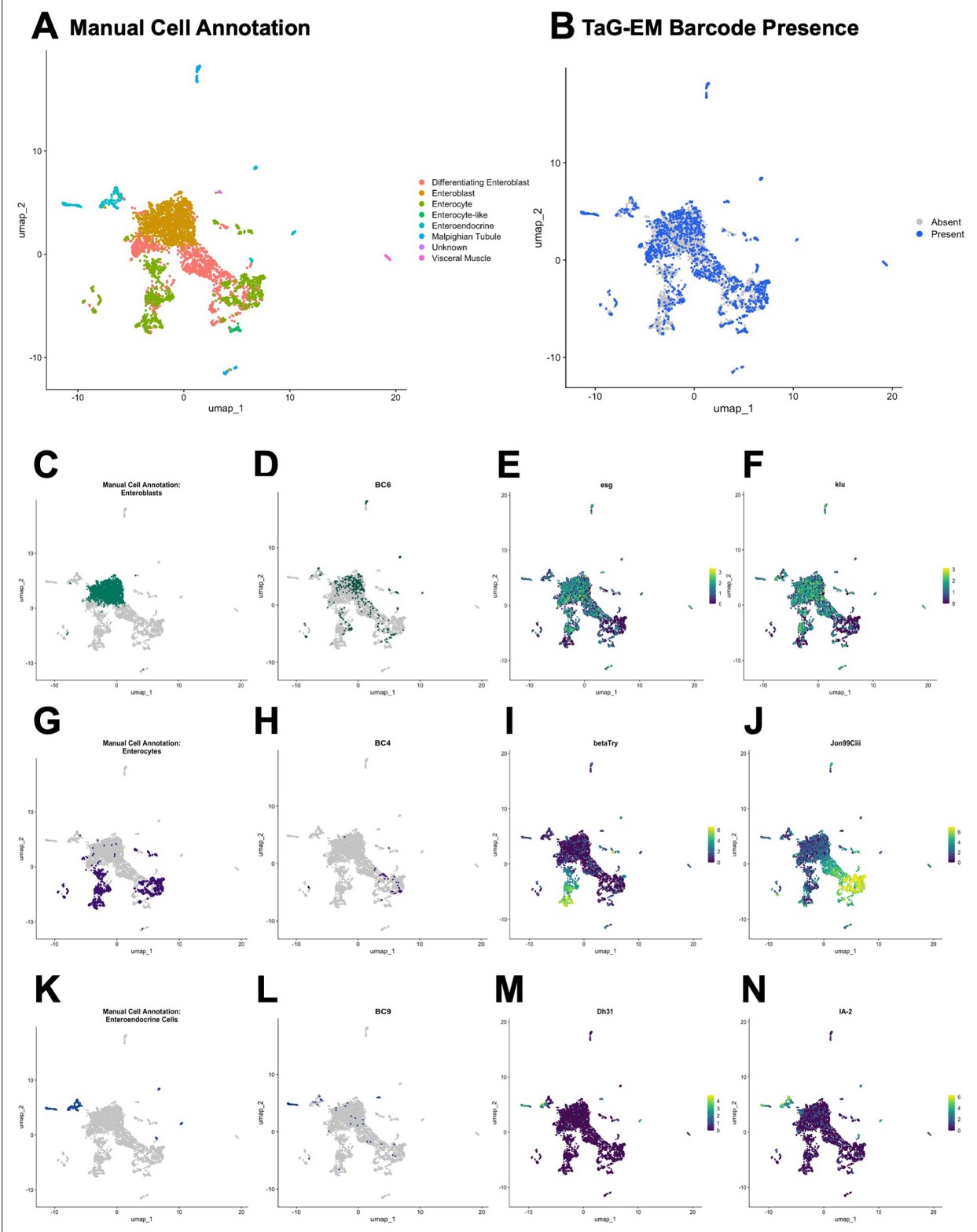

**Figure 6.** Expression of TaG-EM genetic barcodes in larval intestinal cell types. (**A**) UMAP plot of *Drosophila* larval gut cell types. (**B**) Annotation of cells associated with a TaG-EM barcode across all 8 multiplexed experimental conditions using data from the gene expression library and an enriched TaG-EM barcode library. (**C**) Annotated enteroblast cells. (**D**) Presence of TaG-EM barcode (BC6) driven by the EB-Gal4 line using data from the gene expression library and an enriched TaG-EM barcode library. Gene expression levels of enteroblast marker genes (**E**) *esg*, (**F**) *klu*. (**G**) Annotated enterocyte cells. (**H**) Presence of TaG-EM barcode (BC4) driven by the EC-Gal4 line using data from the gene expression library and an enriched TaG-EM

*Figure 6 continued on next page*

barcode library. Gene expression levels of enterocyte marker genes (**I**) *betaTry*, (**J**) *Jon99Ciii*. (**K**) Annotated enteroendocrine cells. (**L**) Presence of TaG-EM barcode (BC9) driven by the EE-Gal4 line using data from the gene expression library and an enriched TaG-EM barcode library. Gene expression levels of enteroendocrine cell marker genes (**M**) *Dh31*, (**N**) *IA-2*.

The online version of this article includes the following figure supplement(s) for figure 6:

**Figure supplement 1.** Dissociated intestinal cell viability.

**Figure supplement 2.** BD FACSDiva 8.0.1 gating for sorted cells.

**Figure supplement 3.** Expression of TaG-EM genetic barcodes in larval intestinal precursor cells.

**Figure supplement 4.** BD FACSDiva 8.0.1 gating for sorted cells.

**Figure supplement 5.** TaG-EM-based doublet identification.

**Figure supplement 6.** Clustering and automated annotation.

**Figure supplement 7.** Expression of TaG-EM genetic barcodes in larval intestinal cell types.

**Figure supplement 8.** Optimizing amplification of the TaG-EM barcode library.

**Figure supplement 9.** Performance of the enriched TaG-EM barcode library.

**Figure supplement 10.** Expression of the PMG-Gal4-driven TaG-EM barcodes.

**Figure supplement 11.** Characterization of Gal4 line expression in the larval gut.

indicated that although this line labeled a high percentage of enterocytes, expression was restricted to an area at the anterior and middle of the midgut, with gaps between these regions and at the posterior (*Figure 6—figure supplement 11*). This could explain the absence of subsets of enterocytes, such as those labeled by *betaTry*, which exhibits regional expression in R2 of the adult midgut (*Buchon et al., 2013*).

Finally, expression of TaG-EM barcode 9, which was expressed using the EE-Gal4 driver line, was observed in clusters annotated as enteroendocrine cells (*Figure 6K–L*) and that also expressed enteroendocrine cell derived neuropeptide genes such as *Dh31* (*Figure 6M*) and other enteroendocrine markers such as *IA-2*, a tyrosine phosphatase involved in the secretion of insulin-like peptide (*Figure 6N*). Detailed characterization of the EE-Gal4 driver line indicated that ~80–85% of Prospero-positive enteroendocrine cells are labeled in the anterior and middle of the larval midgut, with a lower percentage (~65%) of Prospero-positive cells labeled in the posterior midgut (*Figure 6—figure supplement 11*). As with the enterocyte labeling, and consistent with the Gal4 driver expression pattern, the EE-Gal4 expressed TaG-EM barcode 9 did not label all classes of enteroendocrine cells and other clusters of presumptive enteroendocrine cells expressing other neuropeptides such as *Orcokinin*, *AstA*, and *AstC*, or neuropeptide receptors such as *CCHa2* (not shown) were also observed. The EE-Gal4 driver uses *Dh31* regulatory elements, so it is not surprising that the TaG-EM barcodes specifically labeled *Dh31*-positive enteroendocrine cells and this result further highlights the ability to target specific genetically defined cell types using TaG-EM based on in vivo cell labeling. Taken together, these results demonstrate that TaG-EM can be used to label specific cell populations that correlate with Gal4-driven expression patterns in vivo for subsequent identification in single-cell sequencing data.

## Discussion

Advances in next-generation sequencing, as well as single-cell and spatial genomics are enabling new types of detailed analyses to study important biological processes such as development and nervous system function. Here, we describe TaG-EM, a genetic barcoding strategy that enables novel capabilities in several different experimental contexts (*Figure 1*).

We demonstrate that the genetic barcodes can be quantified from mixtures of barcoded fly lines using next-generation sequencing. Analysis of structured pools of flies with defined inputs suggests that TaG-EM barcode measurements are highly accurate and reproducible, particularly in cases where multiple barcodes are used to label an experimental condition and averaged (*Figure 2*). Sequencing-based TaG-EM measurements recapitulated more laborious, one-at-a-time measurements in a phototaxis assay, an age-dependent fecundity assay, and a gut motility assay, demonstrating that TaG-EM can be used to measure behavior or other phenotypes in multiplexed, pooled populations

(*Figures 3–4*). We did note that one line (TaG-EM barcode 14) exhibited poor performance in oviposition assays, suggesting that barcode performance should be verified for a specific assay. We excluded this poor performing barcode line from the fecundity tests; however, backcrossing is often used to bring reagents into a consistent genetic background for behavioral experiments and could also potentially be used to address behavior-specific issues with specific TaG-EM lines. In addition, other strategies such as averaging across multiple barcode lines (3–4 per condition, which yielded R² values >0.99 in tests with structured pools) or permutation of barcode assignment across replicates could also mitigate such deficiencies. Currently, up to 176 conditions can be multiplexed in a single pooled experiment with existing TaG-EM lines, but because sequencing indices can be added after amplification in a separate indexing PCR step, many hundreds or even thousands of such experiments can be multiplexed in a single-sequencing run. While the utility of TaG-EM barcode-based quantification will vary based on the number of conditions being analyzed and the ease of quantifying the behavior or phenotype by other means, we demonstrate that TaG-EM can be employed to cost-effectively streamline labor-intensive assays and to quantify phenotypes with small effect sizes (*Figure 4*, *Figure 4—figure supplement 2*). An additional benefit of multiplexed TaG-EM behavioral measurements is that the experimental conditions are effectively blinded as the multiplexed conditions are intermingled in a single assay.

In addition, we show that TaG-EM barcodes can be expressed by tissue-specific Gal4 drivers and used to tag specific cell populations upstream of single-cell sequencing (*Figures 5–6*). This capability will allow for positive identification of cell clusters in cell atlas projects and will facilitate multiplexing of single-cell sequencing experiments. Recently, a conceptually similar approach called RABID-Seq (*Clark et al., 2021*) has been described, which allows trans-synaptic labeling of neural circuits using barcoded viral transcripts. However, one distinction between the two approaches is that RABID-Seq relies on stochastic viral infection of mammalian cells while TaG-EM allows reproducible targeting of defined cell populations allowing unambiguous cell identification and potentially allowing the same cell populations to be assessed at different timepoints or in the context of different experimental manipulations. One current limitation is that TaG-EM barcodes are not observed in all cells in single-cell gene expression data. It is likely that the strength of the Gal4 driver contributes to the labeling density. However, we also observed variable recovery of TaG-EM barcodes that were all driven by the same pan-midgut Gal4 driver (*Figure 6—figure supplement 10*). For single-cell RNA-Seq experiments, the cost savings of multiplexing is roughly the cost of a run divided by the number of independent lines multiplexed, plus labor savings by also being able to multiplex upstream flow cytometry, minus loss of unbarcoded cells. Our experiments indicated that for the specific drivers we tested TaG-EM barcodes are detected in around one quarter of the cells if relying on endogenous expression in the gene expression library, though this fraction was higher (~37%) if sequencing an enriched TaG-EM barcode library in parallel (*Figure 6*, *Figure 6—figure supplement 8* and *Figure 6—figure supplement 9*).

In the future, generation of additional TaG-EM lines will enable higher levels of multiplexing. In addition, while the original TaG-EM lines were made using a membrane-localized myr::GFP construct, variants that express GFP in other cell compartments such as the cytoplasm or nucleus could be constructed to enable increased expression levels or purification of nuclei. Nuclear labeling could also be achieved by co-expressing a nuclear GFP construct with existing TaG-EM lines in analogy to the use of hexameric GFP described above.

In summary, combined with the large collections of Gal4 and split-Gal4 lines that have been established in *Drosophila* that enable precise targeting of a high proportion of cell types (*Ariyapala et al., 2020*; *Aso et al., 2014*; *Davis et al., 2020*; *Gohl et al., 2011*; *Kanca et al., 2022*; *Namiki et al., 2018*; *Pfeiffer et al., 2010*; *Pfeiffer et al., 2008*; *Venken et al., 2011*; *Zirin et al., 2024*), TaG-EM provides a means to target and label cells in vivo for subsequent detection in single-cell sequencing. Moreover, these genetic barcodes can be used to multiplex behavioral or other phenotypic measurements. Thus, TaG-EM provides a flexible system for barcoding cells and organisms.

# Methods

## Key resources table

| Reagent type (species) or resource | Designation | Source or reference | Identifiers | Additional information |
|---|---|---|---|---|
| Genetic reagent (*D. melanogaster*) | IsoD1 | Clandinin Lab, Stanford University, *Silies et al., 2013* | Wild type | |
| Genetic reagent (*D. melanogaster*) | w-;+;+ (IsoD1) | Clandinin Lab, Stanford University, *Silies et al., 2013* | | |
| Genetic reagent (*D. melanogaster*) | atttP2 line | Transgenic RNAi Project | RRID:BDSC_25710 | P{y[+t7.7]=nanos-phiC31\int.NLS}X, y (*Alegria et al., 2024*) sc (*Alegria et al., 2024*) v (*Alegria et al., 2024*) sev (*Ingham, 1988*); P{y[+t7.7]=CaryP}attP2 |
| Genetic reagent (*D. melanogaster*) | norpA | William Pak, Purdue University, West Lafayette | RRID:BDSC_9048 | w[*] norpA[P24] |
| Genetic reagent (*D. melanogaster*) | UAS-myr::GFP(pJFRC12) | Gerald M. Rubin & Barret Pfeiffer, Howard Hughes Medical Institute, Janelia Research Campus | RRID:BDSC_32197 | w[*]; P{y[+t7.7] w[+mC]=10XUAS-IVS-myr::GFP}attP2 |
| Genetic reagent (*D. melanogaster*) | Hexameric GFP lines | Nicholas Sokol, Indiana University, Bloomington | RRID:BDSC_91402, RRID:BDSC_91403 | w[*]; P{y[+t7.7] w[+mC]=R57 F07-p65.AD.A}attP40; P{y[+t7.7] w[+mC]=UAS-DSCP-6XEGFP}attP2 w[*]; PBac{y[+mDint2] w[+mC]=UAS-DSCP-6XEGFP}VK00018; P{y[+t7.7] w[+mC]=R57 F07-GAL4.DBD.A}attP2/TM6C, Sb (*Alegria et al., 2024*) Tb (*Alegria et al., 2024*) |
| Genetic reagent (*D. melanogaster*) | UAS-6XmCherry-HA | Steve Stowers, Montana State University | RRID:BDSC_52268 | y (*Alegria et al., 2024*) w[*]; wg[Sp-1]/CyO, P{Wee-P.ph0}Bacc[Wee-P20]; P{y[+t7.7] w[+mC]=20XUAS-6XmCherry-HA}attP2 |
| Genetic reagent (*D. melanogaster*) | UAS-GFP.nls | Bruce Edgar, Fred Hutchinson Cancer Center | RRID:BDSC_4776 | w[1118]; P{w[+mC]=UAS GFP.nls}8 |
| Genetic reagent (*D. melanogaster*) | esg-GFP.FPTB | modERN Project | RRID:BDSC_83386 | y (*Alegria et al., 2024*) w[*]; PBac{y[+mDint2] w[+mC]=esg GFP.FPTB}VK00031 |
| Genetic reagent (*D. melanogaster*) | dpp-Gal4 driver | Karen Staehling-Hampton, University of Wisconsin, Madison | RRID:BDSC_1553 | w[*]; wg[Sp-1]/CyO; P{w[+mW.hs]=GAL4 dpp.blk1}40 C.6/TM6B, Tb (*Alegria et al., 2024*) |
| Genetic reagent (*D. melanogaster*) | Act-Gal4 driver | Yash Hiromi, National Institute of Genetics | RRID:BDSC_4414 | y (*Alegria et al., 2024*) w[*]; P{w[+mC]=Act5 C-GAL4}25FO1/CyO, y[+] |
| Genetic reagent (*D. melanogaster*) | Tub-Gal4 driver | Liqun Luo, Stanford University | RRID:BDSC_5138 | y (*Alegria et al., 2024*) w[*]; P{w[+mC]=tubP-GAL4}LL7/TM3, Sb (*Alegria et al., 2024*) Ser (*Alegria et al., 2024*) |
| Genetic reagent (*D. melanogaster*) | Mhc-Gal4 driver | Frank Schnorrer, Max Planck Institute of Biochemistry | RRID:BDSC_55132 | P{w[+mC]=Mhc-GAL4.K}1, w[*]/FM7c |
| Genetic reagent (*D. melanogaster*) | PC-Gal4 driver lines | Barry Dickson, Howard Hughes Medical Institute, Janelia Research Campus | RRID:BDSC_73356 RRID:BDSC_75528 | w[1118]; P{y[+t7.7] w[+mC]=VT004241 p65.AD}attP40 w[1118]; P{y[+t7.7] w[+mC]=VT024642 GAL4.DBD}attP2 |
| Genetic reagent (*D. melanogaster*) | PC-Gal4 driver (with UAS-Stinger) lines | Nicholas Sokol, Indiana University, Bloomington | RRID:BDSC_91400 RRID:BDSC_91401 | w[*]; P{y[+t7.7] w[+mC]=VT004241 p65.AD}attP40, P{w[+mC]=UAS-Stinger}2/CyO; l(3)*[*]/ TM3, Sb (*Alegria et al., 2024*) Ser (*Alegria et al., 2024*) w[*]; P{y[+t7.7] w[+mC]=VT024642 GAL4.DBD}attP2, P{w[+mC]=UAS-Stinger}3 |
| Genetic reagent (*D. melanogaster*) | EC-Gal4 driver | Nicholas Sokol, Indiana University, Bloomington | RRID:BDSC_91406 | w[*]; P{y[+t7.7] w[+mC]=CG10116 GAL4.DBD}su(Hw)attP6, P{y[+t7.7] w[+mC]=VT004958 p65.AD}attP40/CyO |
| Genetic reagent (*D. melanogaster*) | EB-Gal4 driver lines | Nicholas Sokol, Indiana University, Bloomington | RRID:BDSC_91398 RRID:BDSC_91404 | w[*]; P{y[+t7.7] w[+mC]=CG10116 p65.AD}attP40 w[*]; P{y[+t7.7] w[+mC]=Su(H)GBE-GAL4.DBD}attP2/TM6B, Tb[+] |
| Genetic reagent (*D. melanogaster*) | EE-Gal4 driver lines | Nicholas Sokol, Indiana University, Bloomington | RRID:BDSC_91402 RRID:BDSC_91403 | w[*]; P{y[+t7.7] w[+mC]=R57 F07-p65.AD.A}attP40; P{y[+t7.7] w[+mC]=UAS-DSCP-6XEGFP}attP2 w[*]; PBac{y[+mDint2] w[+mC]=UAS-DSCP-6XEGFP}VK00018; P{y[+t7.7] w[+mC]=R57 F07-GAL4.DBD.A}attP2/TM6C, Sb (*Alegria et al., 2024*) Tb (*Alegria et al., 2024*) |
| Genetic reagent (*D. melanogaster*) | PMG-Gal4 driver lines | Nicholas Sokol, Indiana University, Bloomington | RRID:BDSC_91398 RRID:BDSC_91399 | w[*]; P{y[+t7.7] w[+mC]=CG10116 p65.AD}attP40 w[*]; P{y[+t7.7] w[+mC]=CG10116 GAL4.DBD}su(Hw)attP6 |
| Genetic reagent (*D. melanogaster*) | TaG-EM lines | This study, *Alegria et al., 2024* | | Available upon request |

*Continued on next page*

*Continued*

| Reagent type (species) or resource | Designation | Source or reference | Identifiers | Additional information |
|---|---|---|---|---|
| Genetic reagent (*D. melanogaster*) | TaG-EM lines +6 xGFP (x20) | This study | RRID:BDSC_99608<br>RRID:BDSC_99609<br>RRID:BDSC_99610<br>RRID:BDSC_99611<br>RRID:BDSC_99612<br>RRID:BDSC_99613<br>RRID:BDSC_99614<br>RRID:BDSC_99615<br>RRID:BDSC_99616<br>RRID:BDSC_99617<br>RRID:BDSC_99618<br>RRID:BDSC_99619<br>RRID:BDSC_99620<br>RRID:BDSC_99621<br>RRID:BDSC_99622<br>RRID:BDSC_99623<br>RRID:BDSC_99624<br>RRID:BDSC_99625<br>RRID:BDSC_99626<br>RRID:BDSC_99627 | These lines are available from the Bloomington *Drosophila* Stock Center (stock numbers 99608–99627) |
| Antibody | Anti-GFP (rabbit polyclonal) | ThermoFisher | A-6455<br>RRID:AB_221570 | 1:1000 dilution |
| Antibody | Anti-mCherry (mouse monoclonal) | DSHB | 3A11<br>RRID:AB_2617430 | 1:20 dilution |
| Antibody | Anti-Prospero (mouse monoclonal) | DSHB | MR1A<br>RRID:AB_528440 | 1:50 dilution |
| Antibody | Anti-Pdm1 (mouse monoclonal) | DSHB | Nub2D4<br>RRID:AB_2722119 | 1:30 dilution |
| Antibody | Alexa Fluor 647 Goat Anti-mouse conjugated antibody (goat polyclonal) | ThermoFisher | A-21236<br>RRID:AB_2535805 | 1:200 dilution |
| Antibody | Alexa Fluor 488 Goat Anti-rabbit IgG conjugated antibody (goat polyclonal) | ThermoFisher | A-11008<br>RRID:AB_143165 | 1:200 dilution |
| Recombinant DNA reagent | pJFRC12-10XUAS-IVS-myr::GFP plasmid | Gerald Rubin Lab | RRID:Addgene_26222 | Addgene Plasmid #26222 |
| sequence-based reagent | TaG-Me construct gBlock | Integrated DNA Technologies (IDT) | | caaaggaaaaagctgcactgctataca agaaaattatggaaaaatatttgatgtat agtgccttgactagagatcataatcagc cataccacatttgtagaggttttacttgcttt aaaaaacctcccacacctccccctgaac ctgaaacataaaatgaatgc aattgttgtt gttaacttgtttattgcagcttataa CTTCCAACAACCGGAAGTGA NNNNNNNNNNNNNNNtggttaca aataaagcaatagcatcacaaatttcaca aataaagcatttttttca ctgcattctagtt gtggtttgtccaaactcatcaatgt atcttatcatgtctggatcgatctggccgg ccgtttaaacgaat tcttgaagacgaaag ggcctcgtgatacgcctattttttataggttaa tgtcatgataataatg |
| Sequence-based reagent | SV40_post_R | IDT | | GCCAGATCGATCCAGACATGA |
| Sequence-based reagent | SV40_5 F | IDT | | CTCCCCCTGAACCTGAAACA |
| Sequence-based reagent | B2_3'F1_Nextera | IDT | | TCGTCGGCAGCGTCAGATGT GTATAAGAGACAGCTTCCAACAACCGGAAG*TGA |
| Sequence-based reagent | B2_3'F1_Nextera_2 | IDT | | TCGTCGGCAGCGTCAGATGT GTATAAGAGACAGAGCTTCCAACAACCGGAAG*TGA |
| Sequence-based reagent | B2_3'F1_Nextera_4 | IDT | | TCGTCGGCAGCGTCAGATGT GTATAAGAGACAGTCGACTTCCAACAACCGGAAG*TGA |
| Sequence-based reagent | B2_3'F1_Nextera_6 | IDT | | TCGTCGGCAGCGTCAGATGT GTATAAGAGACAGGAAGAGCTTCCAACAACCGGAAG*TGA |
| Sequence-based reagent | SV40_pre_R_Nextera | IDT | | GTCTCGTGGGCTCGGAGATGT GTATAAGAGACAGATTTGTGAAATTTGTGATGCTATTGC*T TT |
| Sequence-based reagent | SV40_post_R_Nextera | IDT | | GTCTCGTGGGCTCGGAGATGT GTATAAGAGACAGGCCAGATCGATCCAGACA*TGA |
| Sequence-based reagent | Forward indexing primer | IDT | | AATGATACGGCGACCACCGAGA TCTACACXXXXXXXXTCGTCGGCAGCGTC |

*Continued on next page*

*Continued*

| Reagent type (species) or resource | Designation | Source or reference | Identifiers | Additional information |
|---|---|---|---|---|
| Sequence-based reagent | Reverse indexing primer | IDT | | CAAGCAGAAGACGGCATACGAGA TXXXXXXXXGTCTCGTGGGCTCGG |
| Sequence-based reagent | UMGC_IL_TaGEM_SpikeIn_v1 | IDT | | GTGACTGGAGTTCAGACGTGTGCT CTTCCGATCTCTTCCAACAACCGGAAGT*G*A |
| Sequence-based reagent | UMGC_IL_TaGEM_SpikeIn_v2 | IDT | | GTGACTGGAGTTCAGACGTGTGCTCT TCCGATCTGCAGCTTATAACTTCCAACAACCGGAAGT*G*A |
| Sequence-based reagent | UMGC_IL_TaGEM_SpikeIn_v3 | IDT | | TGTGCTCTTCCGATCTGCAGCTTA TAACTTCCAACAACCGGAAGT*G*A |
| Sequence-based reagent | D701_TaGEM | IDT | | CAAGCAGAAGACGGCATACGAGAT CGAGTAATGTGACTGGAGTTCAGAC GTGTGCTCTTC CGATCTGCAGC*T*T |
| Sequence-based reagent | SI PCR Primer | IDT | | AATGATACGGCGACCACCGAGATCT ACACTCTTTCCCTACACGACGC*T*C |
| Sequence-based reagent | UMGC_IL_DoubleNest | IDT | | GTGACTGGAGTTCAGACGTGTGCT CTTCCGATCTGCAGCTTATAACTTCC AACAACCGG*A* A |
| Sequence-based reagent | P5 | IDT | | AATGATACGGCGACCACCGA |
| Sequence-based reagent | D701 | IDT | | GATCGGAAGAGCACACGTCTGAACTC CAGTCACATTACTCGATCTCGTATGCC GTCTTCTG CTTG |
| Sequence-based reagent | D702 | IDT | | GATCGGAAGAGCACACGTCTGAACT CCAGTCACTCCGGAGAATCTCGTATG CCGTCTTCT GCTTG |
| Commercial assay or kit | QIAprep Spin MiniPrep kit | Qiagen | 27104 | |
| Commercial assay or kit | *ApaLI* restriction enzyme | New England BioLabs (NEB) | R0507S | |
| Commercial assay or kit | *PsiI* restriction enzyme | NEB | R0657 | |
| Commercial assay or kit | *EcoRI* restriction enzyme | NEB | R0101S | |
| Commercial assay or kit | Cutsmart Buffer | NEB | B6004S | |
| Commercial assay or kit | Calf Intestinal Phosphatase (CIP) | NEB | M0290S | |
| Commercial assay or kit | T4 DNA ligase | NEB | M0202S | |
| Commercial assay or kit | TOP10 competent cells | Invitrogen | C404010 | |
| Commercial assay or kit | QIAquick Gel Purification Kit | Qiagen | 28104 | |
| Commercial assay or kit | Quant-iT PicoGreen dsDNA assay | ThermoFisher | P11496 | |
| Commercial assay or kit | GeneJET genomic DNA purification Kit | ThermoFisher | K0701 | |
| Commercial assay or kit | *Taq* DNA Polymerase | Qiagen | 201203 | |
| Commercial assay or kit | Exo-CIP Rapid PCR Cleanup Kit | NEB | E1050S | |
| Commercial assay or kit | Q5 High-Fidelity DNA Polymerase | NEB | M0491S | |
| Commercial assay or kit | KAPA HiFi HotStart ReadyMix | Roche | KK2601 | Material Number: 07958927001 |
| Commercial assay or kit | SequalPrep Normalization Plate Kit, 96-well | ThermoFisher | A1051001 | |

*Continued on next page*

*Continued*

| Reagent type (species) or resource | Designation | Source or reference | Identifiers | Additional information |
|---|---|---|---|---|
| Commercial assay or kit | Qubit dsDNA high sensitivity assay | ThermoFisher | Q32851 | |
| Commercial assay or kit | Chromium Next GEM Single Cell 3' Kit v3.1, 4 rxns | 10x Genomics | PN-1000269 | |
| Commercial assay or kit | Chromium Next GEM Chip G Single Cell Kit, 16 rxns | 10x Genomics | PN-1000127 | |
| Commercial assay or kit | Dual Index Kit TT Set A, 96 rxns | 10x Genomics | PN-1000215 | |
| Chemical compound, drug | Ampicillin | Sigma | A9518-5G | |
| Chemical compound, drug | AMPure XP beads | Beckman Coulter | A63881 | |
| Chemical compound, drug | D-(+)-Glucose | Sigma-Aldrich | G7021 | |
| Chemical compound, drug | Caffeine | Sigma-Aldrich | W222402 | |
| Chemical compound, drug | Normal Goat Serum | Abcam | ab7481 | |
| Chemical compound, drug | 1xPBS | Corning | 21040CV | |
| Chemical compound, drug | paraformaldehyde | Electron Microscopy Sciences | 15714 | |
| Chemical compound, drug | Triton X-100 | Sigma-Aldrich | X100-5ML | |
| Chemical compound, drug | DAPI solution | ThermoFisher | 62248 | |
| Chemical compound, drug | Elastase | Sigma-Aldrich | E7885-20MG | |
| Chemical compound, drug | SPRIselect | Beckman Coulter | B23318 | |
| Software, algorithm | Photo Booth | Apple | | |
| Software, algorithm | Fiji | *Schindelin et al., 2012* | RRID:SCR_002285 | http://fiji.sc |
| Software, algorithm | R | R Project for Statistical Computing | RRID:SCR_001905 | https://www.r-project.org/ |
| Software, algorithm | Python | Python Programming Language | RRID:SCR_008394 | http://www.python.org/ |
| Software, algorithm | BioPython | *Cock et al., 2009* | RRID:SCR_007173 | http://biopython.org |
| Software, algorithm | Cell Ranger | 10x Genomics | RRID:SCR_017344 | |
| Software, algorithm | cutadapt | *Martin, 2011* | RRID:SCR_011841 | https://cutadapt.readthedocs.io/en/stable/ |
| Software, algorithm | Seurat | *Satija et al., 2015* | RRID:SCR_016341 | https://satijalab.org/seurat/get_started.html |
| Software, algorithm | DecontX | *Yang et al., 2020* | | https://github.com/campbio/celda |
| Software, algorithm | DoubletFinder | *McGinnis et al., 2019* | RRID:SCR_018771 | https://github.com/chris-mcginnis-ucsf/DoubletFinder |
| Software, algorithm | Clustree | *Zappia and Oshlack, 2018* | RRID:SCR_016293 | https://CRAN.R-project.org/package=clustree |
| Software, algorithm | SingleR | *Aran et al., 2019* | RRID:SCR_023120 | https://www.bioconductor.org/packages/release/bioc/html/SingleR.html |
| Other | LED Strip Light Diffusers | Muzata | HSL-0055 | U1SW WW 1 M, LU1 |
| Other | LED Strip Light, White | LEDJUMP | LJSP-111 | Size 2835, 6000 Kelvin color temperature |
| Other | Arduino Uno Rev 3 | Vilros | ARD_A000066 | See 'Phototaxis experiments' in Methods section. |
| Other | Acoustic Foam Panels | ALPOWL | | 1"x12"x12". See 'Phototaxis experiments' in Methods section. |
| Other | 1080 P Day/Night Vision USB Camera, 2MP Infrared Webcam with Automatic IR-Cut Switching and IR LEDs | Arducam | B0506 | See 'Phototaxis experiments' in Methods section. |

*Continued on next page*

*Continued*

| Reagent type (species) or resource | Designation | Source or reference | Identifiers | Additional information |
|---|---|---|---|---|
| Other | AX R confocal microscope | Nikon | | See 'Dissection and immunostaining' in Methods section. |
| Other | FlowMi 40 µM tip filter | Bel-Art | H13680-0040 | See 'Cell dissociation and isolation' in Methods section. |
| Other | LUNA-FL Dual Fluorescence Cell Counter | Logos Biosystems | L20001 | See 'Cell dissociation and isolation' in Methods section. |
| Other | AO/PI dye | Logos Biosystems | F23001 | See 'Cell dissociation and isolation' in Methods section. |
| Other | FACSAria II Cell Sorter | BD Biosciences | | See 'Cell dissociation and isolation' in Methods section. |

## *Drosophila* stocks and maintenance

*Drosophila* stocks were grown at 22 °C on cornmeal agar unless otherwise indicated. The stocks used in this study are described in the Key Resources Table.

## Design and cloning of TaG-EM constructs

A gBlock with the following sequence containing a part of the SV40 3' UTR with a PCR handle (upper-case, below) and a 14 bp randomer sequence just upstream of the SV40 polyadenylation site (bold and underlined, below) was synthesized (Integrated DNA Technologies, IDT): caaaggaaaaagctgcactg ctatacaagaaaattatggaaaaatatttgatgtatagtgccttgactagagatcataatcagccataccacatttgtagaggtttacttg ctttaaaaaacctcccacacctcccctgaacctgaaacataaaatgaatgcaattgttgttgttaacttgtttattgcagcttataaCTTC CAACAACCGGAAGTGANNNNNNNNNNNNNNtggttaca<u>aataaag</u>caatagcatcacaaatttcaca<u>aataaag</u>cat tttttcactg<u>c</u>attctagttgtggtttgtccaaactcatcaatgtatcttatcatgtctggatcgatctggccggccgtttaaacgaattcttgaa gacgaaagggcctcgtgatacgcctattttttataggttaatgtcatgataataatgt.

The gBlock was resuspended in 20 µl EB, incubated at 50 °C for 20 min and then cut with *Psi*I and *Eco*RI (New England Biolabs, NEB) using the following reaction conditions: 4 µl gBlock DNA (35 ng), 2 µl 10 x CutSmart buffer (NEB), 1 µl *Eco*RI enzyme (NEB), 1 µl *Psi*I enzyme (NEB), and 12 µl nuclease-free water were mixed and incubated at 37 °C for 1 hr followed by 65 °C for 20 min to heat inactivate the restriction enzymes. pJFRC12-10XUAS-IVS-myr::GFP plasmid (Addgene, Plasmid #26222; *Pfeiffer et al., 2010*) was digested with the following reaction conditions: 5 µl pJFRC12-10XUAS-IVS-myr::GFP plasmid DNA (~3 µg), 5 µl 10 x CutSmart buffer (NEB), 1 µl *Psi*I enzyme (NEB), 1 µl *Eco*RI enzyme (NEB), and 38 µl nuclease-free water, were mixed and incubated at 37 °C for 1 hr, followed by addition of 1 µl of CIP and incubation for an additional 30 min. The digested vector backbone was gel purified using the QiaQuick Gel Purification Kit (QIAGEN). The digested gBlock was ligated into the digested pJFRC12-10XUAS-IVS-myr::GFP backbone using the following reactions conditions: 4 µl T4 ligase buffer (10 x; NEB), 20 µl plasmid backbone DNA (0.005 pmol), 5 µl gBlock digest DNA (0.03 pmol), 2 µl of T4 DNA ligase (NEB), and 9 µl nuclease-free water were mixed and incubated at 22 °C for 2 hr. 2 µl of the ligation reaction was transformed into 50 µl of TOP10 competent cells (Invitrogen), and the cells were incubated on ice for 30 min, then heat shocked at 42 °C for 30 s, and incubated on ice for 5 min. 250 µl SOC was added and the cells were plated on LB +ampicillin plates and incubated over-night at 37 °C. DNA was isolated from 36 pJFRC12-gBlock colonies using a QIAprep Spin MiniPrep kit (QIAGEN). Expected construct size was verified by diagnostic digest with *Eco*RI and *Apa*LI. DNA concentration was determined using a Quant-iT PicoGreen dsDNA assay (Thermo Fisher Scientific) and the randomer barcode for each of the constructs was determined by Sanger sequencing using the following primers:

SV40_post_R: GCCAGATCGATCCAGACATGA
SV40_5F: CTCCCCCTGAACCTGAAACA

## Generation of TaG-EM transgenic lines

29 sequence verified constructs were normalized, pooled evenly, and injected as a pool into embryos (*Rubin and Spradling, 1982*) expressing PhiC31 integrase and the carrying the *attP2* landing site (BDSC #25710). Injected flies were outcrossed to w- flies, and up to three white +progeny per cross were identified, and the transgenic lines were homozygosed. DNA was extracted (GeneJET genomic DNA purification Kit, Thermo Scientific) and the region containing the DNA barcode was amplified with the following PCR reaction: 2.5 µl 1:10 diluted template DNA, 2 µl 10 x Reaction Buffer (QIAGEN),

0.2 µl dNTP mix (10 µM), 1 µl 10 µM SV40_5 F primer (10 µM), 1 µl SV40_post_R primer (10 µM), 0.8 µl MgCl$_2$ (3 mM), 0.1 µl Taq polymerase (QIAGEN), 12.4 µl nuclease-free water. Reactions were amplified using the following cycling conditions: 95 °C for 5 min, followed by 30 cycles of 94 °C for 30 s, 55 °C for 30 s, 72 °C for 30 s, followed by 72 °C for 5 min. PCR products were treated with Exo-CIP using the following reaction conditions: 5 µl PCR product, 1 µl Exo-CIP Tube A (NEB), 1 µl Exo-CIP Tube B (NEB) were mixed and incubated at 37 °C for 4 min, followed by 80 °C for 1 min. The barcode sequence for each of the independent transgenic lines was determined by Sanger sequencing using the SV40_5 F and SV40_PostR primers. Transgenic lines containing 20 distinct DNA barcodes were recovered (*Figure 1—figure supplement 1*). An additional 156 TaG-EM barcode lines were isolated and sequence verified as described in a separate publication (*Alegria et al., 2024*).

## Optimizing amplification of TaG-EM barcodes for next-generation sequencing

The following primers were evaluated to amplify the TaG-EM barcodes upstream of NGS:

Forward primer pool: four primers with frameshifting bases to increase library sequence diversity in initial sequencing cycles were normalized to 10 µM and pooled evenly to make a B2_3'F1_Nextera_0–6 primer pool:

B2_3'F1_Nextera:   TCGTCGGCAGCGTCAGATGTGTATAAGAGACAGCTTCCAACAACCGGAAG*TGA

B2_3'F1_Nextera_2
TCGTCGGCAGCGTCAGATGTGTATAAGAGACAGAGCTTCCAACAACCGGAAG*TGA

B2_3'F1_Nextera_4
TCGTCGGCAGCGTCAGATGTGTATAAGAGACAGTCGACTTCCAACAACCGGAAG*TGA

B2_3'F1_Nextera_6
TCGTCGGCAGCGTCAGATGTGTATAAGAGACAGGAAGAGCTTCCAACAACCGGAAG*TGA

The following two reverse primers were tested:

SV40_pre_R_Nextera:      GTCTCGTGGGCTCGGAGATGTGTATAAGAGACAGATTTGTGAAATTTGTGATGCTATTGC*TTT

SV40_post_R_Nextera:     GTCTCGTGGGCTCGGAGATGTGTATAAGAGACAGGCCAGATCGATCCAGACA*TGA

SV40_pre_R_nextera is designed to produce a shorter amplicon (200 bp with Illumina adapters and indices added) and SV40-post_R_Nextera is designed to produce a longer amplicon (290 bp with Illumina adapters and indices added).

An initial test was performed with three different polymerases (NEB Q5, KAPA HiFi, and Qiagen Taq) at two different annealing temperatures and with both the B2_3'F1_Nextera/ SV40_pre_R_Nextera and B2_3'F1_Nextera/ SV40_post_R_Nextera primer sets to determine whether the primers amplify as expected (*Figure 1—figure supplement 1*). Two different samples were tested:

1. Pool of 8 putative transformant samples (pooled 5 µl each of 1:10 diluted sample)
2. OreR (wild type - diluted 1:10)

Set up the following PCR reactions:

### Q5 polymerase

2.5 µl template DNA, 1 µl 10 µM Forward primer (10 µM), 1 µl Reverse primer (10 µM), 10 µl 2 x Q5 Master Mix (NEB), 5.5 µl nuclease-free water. Reactions were amplified using the following cycling conditions: 98 °C for 30 s, followed by 30 cycles of 98 °C for 20 s, 55 °C or 60 °C for 15 s, 72 °C for 30 s, followed by 72 °C for 5 min.

### KAPA HiFi polymerase

2.5 µl template DNA, 1 µl 10 µM Forward primer (10 µM), 1 µl Reverse primer (10 µM), 10 µl 2 x KAPA HiFi ReadyMix (Roche), 5.5 µl nuclease-free water. Reactions were amplified using the following cycling conditions: 95 °C for 5 min, followed by 30 cycles of 98 °C for 20 s, 55 °C or 60 °C for 15 s, 72 °C for 30 s, followed by 72 °C for 5 min.

## Taq polymerase

2.5 µl template DNA, 2 µl 10 x Reaction Buffer (QIAGEN), 0.2 µl dNTP mix (10 µM), 1 µl 10 µM Forward primer (10 µM), 1 µl Reverse primer (10 µM), 0.8 µl $MgCl_2$ (3 mM), 0.1 µl Taq polymerase (QIAGEN), 12.4 µl nuclease-free water. Reactions were amplified using the following cycling conditions: 95 °C for 5 min, followed by 30 cycles of 94 °C for 30 s, 55 °C or 60 °C for 30 s, 72 °C for 30 s, followed by 72 °C for 5 min.

Samples were run on a 2% agarose gel to verify amplification products (*Figure 1—figure supplement 1*).

Next, the TaG-EM barcode lines were pooled in either an even or staggered manner. To optimize reaction conditions for the barcode measurements, either 5 ng or 50 ng of DNA was amplified in triplicate for each pool for either 20, 25, or 30 cycles with either KAPA HiFi using the B2_Nextera_F 0–6 forward primer pool together with either the SV40_pre_R_Nextera or the SV40_post_R_Nextera reverse primer. Next, PCR reactions were diluted 1:100 in nuclease-free water and amplified in the following indexing reactions: 3 µl PCR 1 (1:100 dilution), 1 µl indexing primer 1 (5 µM), 1 µl indexing primer 2 (5 µM), and 5 µl 2 x Q5 master mix. The following indexing primers were used (X indicates the positions of the 8 bp indices):

Forward indexing primer:

AATGATACGGCGACCACCGAGATCTACACXXXXXXXXXTCGTCGGCAGCGTC

Reverse indexing primer:

CAAGCAGAAGACGGCATACGAGATXXXXXXXXXGTCTCGTGGGCTCGG

Reactions were amplified using the following cycling conditions: 98 °C for 30 s, followed by 10 cycles of 98 °C for 20 s, 55 °C for 15 s, 72 °C for 1 min, followed by 72 °C for 5 min. Amplicons were then purified and normalized using a SequalPrep normalization plate (Thermo Fisher Scientific), followed by elution in 20 µl of elution buffer. An even volume of the normalized libraries was pooled and concentrated using 1.8 x AMPure XP beads (Beckman Coulter). Pooled libraries were quantified using a Qubit dsDNA high sensitivity assay (Thermo Fisher Scientific) and libraries were normalized to 2 nM for sequencing on the Illumina MiSeq (see below).

## Structured fly pool experiments

Male or female flies from TaG-EM barcode lines were pooled in either an even or staggered manner (*Figure 2A*). For the even pools, three independently pooled samples were constructed in order to assess sample-to-sample variability. DNA was extracted from these structured pools using a protocol adapted from *Huang et al., 2009* (*Huang et al., 2009*), using homemade SPRI beads (*DeAngelis et al., 1995*) in the last purification step and amplified in triplicate using 2.5 µl template DNA (50 ng), 1 µl 10 µM B2_Nextera_F 0–6 primers (10 µM), 1 µl SV40_pre_R_Nextera (10 µM), 10 µl 2 x KAPA HiFi ReadyMix (Roche), 5.5 µl nuclease-free water. Reactions were amplified using the following cycling conditions: 98 °C for 5 min, followed by 30 cycles of 98 °C for 20 s, 60 °C for 15 s, 72 °C for 30 s, followed by 72 °C for 5 min. Amplicons were indexed, normalized, quantified, and prepared for sequencing as described above.

## Phototaxis experiments

### Video-based measurements

A pair of white LED strip lights with Muzata LED Strip Light Diffusers (U1SW WW 1 M, LU1) were mounted withing a light-tight box and controlled using an Vilros Uno Rev 3 microcontroller. Test tubes containing flies were held in place with Acoustic Foam Panels (1"x12"x12", ALPOWL). Videos and images were acquired using an Arducam 1080 P Day & Night Vision USB Camera with an IR filter and using Photo Booth software (Apple). Wild type and *norpA* flies carrying one of four different TaG-EM barcodes were tested in three independent experimental replicates. 20 male flies of each genotype were transferred into 25 mm x 150 mm glass test tubes, incubated at 34 °C for 10 min and then run in the phototaxis assay, where a light at one end of the chamber was turned on for 30 s. Videos of all tests were recorded through the end of the 30 s light pulse. Videos were independently scored by two observers to determine the number of flies in the light-facing or dark-facing tubes and the results

were averaged. A preference index (P.I.) was calculated using the following formula: [(number of flies in light tube) - (number of flies in dark tube)]/(total number of flies).

## TaG-EM measurements

For TaG-EM barcode-based phototaxis measurements, the following genotypes were consolidated into a single test tube:

## Pool A

norpA/Y;TaG-EM BC4/+
norpA/Y;TaG-EM BC3/+
w-/Y;TaG-EM BC2/+
w-/Y;TaG-EM BC1/+

## Pool B

norpA/Y;TaG-EM BC2/+
norpA/Y;TaG-EM BC1/+
w-/Y;TaG-EM BC4/+
w-/Y;TaG-EM BC3/+

These pools were individually incubated at 34 °C for 10 min and then run in the phototaxis assay. Videos of all tests were recorded and at the end of a 30 s light pulse the two test tubes were quickly separated and capped. Flies in each of these tubes were counted, then DNA was extracted from the flies from the light-facing or dark-facing tubes and amplified using 2.5 µl template DNA (50 ng), 1 µl 10 µM B2_Nextera_F 0–6 primers (10 µM), 1 µl SV40_pre_R_Nextera (10 µM), 10 µl 2 x KAPA HiFi ReadyMix (Roche), 5.5 µl nuclease-free water. Reactions were amplified using the following cycling conditions: 95 °C for 5 min, followed by 30 cycles of 98 °C for 20 s, 60 °C for 15 s, 72 °C for 30 s, followed by 72 °C for 5 min. Amplicons were indexed, normalized, quantified, and prepared for sequencing as described above.

## Oviposition experiments

Newly hatched flies (males and females) from three barcode lines were collected at 1-week intervals during 4 consecutive weeks (12 barcode lines in total). Fresh fly food was provided every 3–4 days. Ten days after the last collection, 10 females from each barcode line were taken and pooled together in a collection cage (10 females x 12 barcode lines = 120 females). The remaining females from each barcode line were separated from the males and put in individual collection cages. Two days later, the experiment started and was run for 3 consecutive days. Each day a 1–1.5 hr pre-collection was followed by a 6 hr collection, both at 25 °C. 100 embryos from each individual collection plate were transferred to new plates and incubated for 2 days at 18 °C. The number of hatched larvae were counted and used to calculate the egg survival rate. The pooled collection plate was also incubated at 18 °C and the next day the embryos were dechorionated and frozen. The 12 individual collection plates were kept at 4 °C and the number of embryos counted in the following days. For the barcode measurements, DNA was extracted from the embryos, and amplified using 2.5 µl template DNA (50 ng), 1 µl 10 µM B2_Nextera_F 0–6 primers (10 µM), 1 µl SV40_pre_R_Nextera (10 µM), 10 µl 2 x KAPA HiFi ReadyMix (Roche), 5.5 µl nuclease-free water. Reactions were amplified using the following cycling conditions: 95 °C for 5 min, followed by 30 cycles of 98 °C for 20 s, 60 °C for 15 s, 72 °C for 30 s, followed by 72 °C for 5 min. Amplicons were indexed, normalized, quantified, and prepared for sequencing as described above.

## Larval gut motility experiments
### Preparing yeast food plates

Yeast agar plates were prepared by making a solution containing 20% Red Star Active Dry Yeast 32oz (Red Star Yeast) and 2.4% Agar Powder/Flakes (Thermo Fisher) and a separate solution containing 20% Glucose (Sigma-Aldrich). Both mixtures were autoclaved with a 45 min liquid cycle and then transferred to a water bath at 55 °C. After cooling to 55 °C, the solutions were combined and mixed,

and approximately 5 mL of the combined solution was transferred into 100x15 mm petri dishes (VWR) in a PCR hood or contamination-free area. For blue-dyed yeast food plates, 0.4% Blue Food Color (McCormick) was added to the yeast solution. For the caffeine assays, 300 µL of a solution of 100 mM 99% pure caffeine (Sigma-Aldrich) was pipetted onto the blue-dyed yeast plate and allowed to absorb into the food during the 90 min starvation period.

### Manual gut motility assay

Third instar *Drosophila* larvae were transferred to empty conical tubes that had been misted with water to prevent the larvae from drying out. After a 90-min starvation period, the larvae were moved from the conical to a blue-dyed yeast plate with or without caffeine and allowed to feed for 60 min. Following the feeding period, the larvae were transferred to an undyed yeast plate. Larvae were scored for the presence or absence of a food bolus every 30 min over a 5 hr period. Up to eight experimental replicates/conditions were scored simultaneously.

### TaG-EM gut motility assay

Third instar larvae were starved and fed blue dye-containing food with or without caffeine as described above. An equal number of larvae from each experimental condition/replicate were transferred to an undyed yeast plate. During the 5 hr observation period, larvae were examined every 30 min and larvae lacking a food bolus were transferred to a microcentrifuge tube labeled for the timepoint. Any larvae that died during the experiment were placed in a separate microcentrifuge tube and any larvae that failed to pass the food bolus were transferred to a microcentrifuge tube at the end of the experiment. DNA was extracted from the larvae in each tube and TaG-EM barcode libraries were prepared and sequenced as described above.

## Dissection and immunostaining

Midguts from third instar larvae of driver lines crossed to UAS-GFP.nls or UAS-mCherry were dissected in 1xPBS and fixed with 4% paraformaldehyde (PFA) overnight at 4 °C. Fixed samples were washed with 0.1% PBTx (1 x PBS +0.1% Triton X-100) three times for 10 min each and blocked in PBTxGS (0.1% PBTx +3% Normal Goat Serum) for 2–4 hr at RT. After blocking, midguts were incubated in primary antibody solution overnight at 4 °C. The next day samples were washed with 0.1% PBTx three times for 20 min each and were incubated in secondary antibody solution for 2–3 hr at RT (protected from light) followed by three washes with 0.1% PBTx for 20 min each. One µg/ml DAPI solution prepared in 0.1% PBTx was added to the sample and incubated for 10 min followed by washing with 0.1% PBTx three times for 10 min each. Finally, samples were mounted on a slide glass with 70% glycerol and imaged using a Nikon AX R confocal microscope. Confocal images were processed using Fiji software.

The primary antibodies used were rabbit anti-GFP (A6455,1:1000 Invitrogen), mouse anti-mCherry (3A11, 1:20 DSHB), mouse anti-Prospero (MR1A, 1:50 DSHB) and mouse anti-Pdm1 (Nub 2D4, 1:30 DSHB). The secondary antibodies used were goat anti-mouse and goat anti-rabbit IgG conjugated to Alexa 647 and Alexa 488 (1:200; Invitrogen), respectively. Five larval gut specimens per Gal4 line were dissected and examined.

## Cell dissociation and isolation

Midguts from 3rd instar larvae were dissected in phosphate-buffered saline (PBS) and transferred to microcentrifuge tubes on ice containing PBS +30% normal goat serum (NGS). After dissection, 150 µL of 2.7 mg/mL elastase was added to each sample tube. The tubes were then incubated at 27 °C for 1 hr. During incubation, samples were mixed by pipetting ~30 times every 15 min to improve elastase dissociation of the cells. Samples were then filtered through a 40 µM FlowMi tip filter (Bel-Art) to reduce debris. Afterwards, the samples were quantified on the LUNA-FL Dual Fluorescence Cell Counter (Logos Biosystems) using 9 µL of sample to 1 µL AO/PI dye to ensure there were enough viable cells for flow sorting.

Once quantified, the samples were brought up to a volume of ~1.1 mL with the PBS +30% NGS solution to facilitate flow sorting. The samples were then fluorescently sorted on a FACSAria II Cell Sorter (BD Biosciences) to isolate GFP + cells. Following sorting, samples were centrifuged at 300 x *g* for 10 min to concentrate the cells. The supernatant was aspirated off until 50 µL cell concentrate remained in each sample. Then, the samples were carefully resuspended using wide bore pipette tips

before being combined into one sample tube. This sample was quantified on the LUNA-FL Dual Fluorescence Cell Counter (Logos Biosystems) as described above. If necessary, cells were centrifuged, concentrated, and re-counted.

## Preparation of single-cell sequencing libraries

The resulting pool was prepared for sequencing following the 10x Genomics Single Cell 3' protocol (version CG000315 Rev C), At step 2.2 of the protocol, cDNA amplification, 1 µl of TaG-EM spike-in primer (10 µM) was added to the reaction to amplify cDNA with the TaG-EM barcode. Gene expression cDNA and TaG-EM cDNA were separated using a double-sided SPRIselect (Beckman Coulter) bead clean up following 10x Genomics Single Cell 3' Feature Barcode protocol, step 2.3 (version CG000317 Rev E). The gene expression cDNA was created into a library following the CG000315 Rev C protocol starting at section 3. Custom nested primers were used for enrichment of TaG-EM barcodes after cDNA creation using PCR.

The following primers were tested (see *Figure 6—figure supplement 8*):

UMGC_IL_TaGEM_SpikeIn_v1: GTGACTGGAGTTCAGACGTGTGCTCTTCCGATCTCTTCCAACAACCGGAAGT*G*A

UMGC_IL_TaGEM_SpikeIn_v2: GTGACTGGAGTTCAGACGTGTGCTCTTCCGATCTGCAGCTTATAACTTCCAACAACCGGAAGT*G*A

UMGC_IL_TaGEM_SpikeIn_v3: TGTGCTCTTCCGATCTGCAGCTTATAACTTCCAACAACCGGAAGT*G*A

D701_TaGEM: CAAGCAGAAGACGGCATACGAGATCGAGTAATGTGACTGGAGTTCAGACGTGTGCTCTTCCGATCTGCAGC*T*T

SI PCR Primer: AATGATACGGCGACCACCGAGATCTACACTCTTTCCCTACACGACGC*T*C

UMGC_IL_DoubleNest: GTGACTGGAGTTCAGACGTGTGCTCTTCCGATCTGCAGCTTATAACTTCCAACAACCGG*A*A

P5: AATGATACGGCGACCACCGA

D701: GATCGGAAGAGCACACGTCTGAACTCCAGTCACATTACTCGATCTCGTATGCCGTCTTCTGCTTG

D702: GATCGGAAGAGCACACGTCTGAACTCCAGTCACTCCGGAGAATCTCGTATGCCGTCTTCTGCTTG

After multiple optimization trials, the following steps yielded ~96% on-target reads for the TaG-EM library (*Figure 6—figure supplement 8*, note that for the enriched barcode data shown in *Figure 6*, *Figure 6—figure supplement 9*, a similar amplification protocol was used TaG-EM barcodes were amplified from the gene expression library cDNA and not the SPRI-selected barcode pool). TaG-EM cDNA was amplified with the following PCR reaction: 5 µl purified TaG-EM cDNA, 50 µl 2 x KAPA HiFi ReadyMix (Roche), 2.5 µl UMGC_IL_DoubleNest primer (10 µM), 2.5 µl SI_PCR primer (10 µM), and 40 µl nuclease-free water. The reaction was amplified using the following cycling conditions: 98 °C for 2 min, followed by 15 cycles of 98 °C for 20 s, 63 °C for 30 s, 72 °C for 20 s, followed by 72 °C for 5 min. After the first PCR, the amplified cDNA was purified with a 1.2 x SPRIselect (Beckman Coulter) bead cleanup with 80% ethanol washes and eluted into 40 µL of nuclease-water. A second round of PCR was run with following reaction: 5 µl purified TaG-EM cDNA, 50 µl 2 x KAPA HiFi ReadyMix (Roche), 2.5 µl D702 primer (10 µM), 2.5 µl p5 Primer (10 µM), and 40 µl nuclease-free water. The reaction was amplified using the following cycling conditions: 98 °C for 2 min, followed by 10 cycles of 98 °C for 20 s, 63 °C for 30 s, 72 °C for 20 s, followed by 72 °C for 5 min. After the second PCR, the amplified cDNA was purified with a 1.2 x SPRIselect (Beckman Coulter) bead cleanup with 80% ethanol washes and eluted into 40 µL of nuclease-water. The resulting 3' gene expression library and TaG-EM enrichment library were sequenced together following Scenario 1 of the BioLegend 'Total-Seq-A Antibodies and Cell Hashing with 10x Single Cell 3' Reagents Kit v3 or v3.1' protocol. Additional sequencing of the enriched TaG-EM library also done following Scenario 2 from the same protocol.

## Sequencing

Libraries for TaG-EM barcode analysis from structured pools or from phototaxis or oviposition experiments were denatured with NaOH and prepared for sequencing according to the protocols described in the Illumina MiSeq Denature and Dilute Libraries Guides. Single-cell libraries were sequenced on the Illumina NextSeq 2000 or Illumina NovaSeq 6000. One of the single-cell enriched TaG-EM barcode libraries was sequenced on an Element Aviti sequencer following the manufacturers loading instructions.

## Data analysis

### Behavioral experiments

Demultiplexed fastq files were generated using bcl2fastq or bcl-convert. TaG-EM barcode data was analyzed using custom R and Python scripts and BioPython (*Cock et al., 2009*). Leading primer sequences were trimmed using cutadapt (*Martin, 2011*) and the first 14 bp of the remaining trimmed read were compared to a barcode reference file, with a maximum of 2 mismatches allowed, using a custom script (TaG-EM_barcode_analysis.py) which is available via Github: https://github.com/daryl-gohl/TaG-EM (copy archived at *Gohl, 2024*).

### Single-cell experiments

Data sets were first mapped and analyzed using the Cell Ranger analysis pipeline (10x Genomics). A custom *Drosophila* genome reference was made by combining the BDGP.28 reference genome assembly and Ensembl gene annotations. Custom gene definitions for each of the TaG-EM barcodes were added to the fasta genome file and .gtf gene annotation file. A Cell Ranger reference package was generated with the Cell Ranger *mkref* command. Subsequent single-cell data analysis was performed using the R package Seurat (*Satija et al., 2015*). Cells expressing less than 200 genes and genes expressed in fewer than three cells were filtered from the expression matrix. Next, percent mitochondrial reads, percent ribosomal reads, cell counts, and cell features were graphed to determine optimal filtering parameters. DecontX (*Yang et al., 2020*) was used to identify empty droplets, to evaluate ambient RNA contamination, and to remove empty cells and cells with high ambient RNA expression. DoubletFinder (*McGinnis et al., 2019*) to identify droplet multiplets and remove cells classified as multiplets. Clustree (*Zappia and Oshlack, 2018*) was used to visualize different clustering resolutions and to determine the optimal clustering resolution for downstream analysis. Finally, SingleR (*Aran et al., 2019*) was used for automated cell annotation with a gut single-cell reference from the Fly Cell Atlas (*Li et al., 2022*). The data set was manually annotated using the expression patterns of marker genes known to be associated with cell types of interest. To correlate TaG-EM barcodes with cell IDs in the enriched TaG-EM barcode library, a custom Python script was used (TaG-EM_barcode_Cell_barcode_correlation.py), which is available via Github: https://github.com/darylgohl/TaG-EM (copy archived at *Gohl, 2024*).

## Acknowledgements

We thank our colleagues in the University of Minnesota Genomics Center (RRID:SCR_012413), in particular Aaron Becker, Dylan Cole, and Logan Silber for help with DNA sequencing, Emma Stanley, Fernanda Rodriguez, and Patrick Grady for assistance and advice on single-cell sequencing, and Thomas Clandinin, Nam Chul Kim, Kenneth Beckman, Andrew Alegria, Troy Louwagie, and Aaron Barnes for helpful feedback and discussions. This work was supported by the resources and staff at the University of Minnesota University Imaging Centers (RRID:SCR_020997). The authors acknowledge the Minnesota Supercomputing Institute (MSI) at the University of Minnesota for providing resources that contributed to the research results reported within this paper. URL: http://www.msi.umn.edu. Stocks obtained from the Bloomington *Drosophila* Stock Center (NIH P40OD018537) were used in this study. Antibodies were obtained from the Developmental Studies Hybridoma Bank (DSHB). Prospero (MR1A) was deposited to the DSHB by C.Q. Doe and Nub 2D4 was deposited to the DSHB by Michalis Averof. This study was supported by a grant from the Winston and Maxine Wallin Neuroscience Discovery Fund.

# Additional information

## Funding

| Funder | Grant reference number | Author |
| --- | --- | --- |
| Winston and Maxine Wallin Neuroscience Discovery Fund | | Daryl M Gohl |

The funders had no role in study design, data collection and interpretation, or the decision to submit the work for publication.

## Author contributions

Jorge Blanco Mendana, Conceptualization, Formal analysis, Investigation, Visualization, Writing – review and editing; Margaret Donovan, Formal analysis, Investigation, Visualization, Writing – review and editing; Lindsey Gengelbach O'Brien, Benjamin Auch, Investigation, Writing – review and editing; John Garbe, Data curation, Formal analysis, Writing – review and editing; Daryl M Gohl, Conceptualization, Data curation, Formal analysis, Supervision, Funding acquisition, Investigation, Visualization, Writing – original draft, Writing – review and editing

## Author ORCIDs

Margaret Donovan ⓘ https://orcid.org/0009-0007-1779-4459
Lindsey Gengelbach O'Brien ⓘ https://orcid.org/0009-0007-5423-7387
Daryl M Gohl ⓘ https://orcid.org/0000-0002-4434-2788

Reviewer #1 (Public review): https://doi.org/10.7554/eLife.88334.3.sa1
Reviewer #2 (Public review): https://doi.org/10.7554/eLife.88334.3.sa2
Author response https://doi.org/10.7554/eLife.88334.3.sa3

# Additional files

## Supplementary files

MDAR checklist

## Data availability

Availability of data, code, and materials Sequencing data for this project is available through the National Center for Biotechnology Information (NCBI) Sequence Read Archive BioProject PRJNA912199. Fly stocks containing 20 of the TaG-EM barcodes together with an additional UAS hexameric GFP expression construct will be available from the Bloomington *Drosophila* Stock Center. Additional TaG-EM barcode stocks are available upon request. Single cell analysis code and the TaG-EM barcode analysis script and barcode reference fasta files are available via Github: https://github.com/darylgohl/TaG-EM (copy archived at *Gohl, 2024*).

The following dataset was generated:

| Author(s) | Year | Dataset title | Dataset URL | Database and Identifier |
| --- | --- | --- | --- | --- |
| Gohl D | 2022 | Deterministic Genetic Barcoding for Multiplexed Behavioral and Single Cell Transcriptomic Studies | https://www.ncbi.nlm.nih.gov/bioproject/PRJNA912199 | NCBI BioProject, PRJNA912199 |

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
