## [Editor Report · eLife Assessment]

This **useful** study presents a genetically encoded barcoding system that could advance transcriptomic studies and that has the potential for further applications, such as in high-throughput population-scale behavioral measurements. The evidence supporting the claims of the authors is **solid** and highlights both the usefulness and the limitations of the approach.

---

## [Referee Report · Reviewer #1 (Public review)]

The aim of this paper is to describe a novel method for genetic labelling of animals or cell populations, using a system of DNA/RNA barcodes.

Strengths:

• The author's attempt at providing a straightforward method for multiplexing *Drosophila* samples prior to scRNA-seq is commendable. The perspective of being able to load multiple samples on a 10X Chromium without antibody labelling is appealing.

• The authors are generally honest about potential issues in their method, and areas that would benefit from future improvement.

• The article reads well. Graphs and figures are clear and easy to understand.

Weaknesses:

• The usefulness of TaG-EM for phototaxis, egg laying or fecundity experiments is questionable. The behaviours presented here are all easily quantifiable, either manually or using automated image-based quantification, even when they include a relatively large number of groups and replicates. Despite their claims (e.g., L311-313), the authors do not present any real evidence about the cost- or time-effectiveness of their method in comparison to existing quantification methods.

• Behavioural assays presented in this article have clear outcomes, with large effect sizes, and therefore do not really challenge the efficiency of TaG-EM. By showing a T-maze in Fig 1B, the authors suggest that their method could be used to quantify more complex behaviours. Not exploring this possibility in this manuscript seems like a missed opportunity.

• Experiments in Figs S3 and S6 suggest that some tags have a detrimental effect on certain behaviours or on GFP expression. Whereas the authors rightly acknowledge these issues, they do not investigate their causes. Unfortunately, this question the overall suitability of TaG-EM, as other barcodes may also affect certain aspects of the animal's physiology or behaviour. Revising barcode design will be crucial to make sure that sequences with potential regulatory function are excluded.

• For their single-cell experiments, the authors have used the 10X Genomics method, which relies on sequencing just a short segment of each transcript (usually 50-250bp - unknown for this study as read length information was not provided) to enable its identification, with the matching paired-end read providing cell barcode and UMI information (Macosko et al., 2015). With average fragment length after tagmentation usually ranging from 300-700bp, a large number of GFP reads will likely not include the 14bp TaG-EM barcode. When a given cell barcode is not associated with any TaG-EM barcode, then demultiplexing is impossible. This is a major problem, which is particularly visible in Figs 5 and S13. In 5F, BC4 is only detected in a couple of dozen cells, even though the Jon99Ciii marker of enterocytes is present in a much larger population (Fig 5C). Therefore, in this particular case, TaG-EM fails to detect most of the GFP-expressing cells. Similarly, in S13, most cells should express one of the four barcodes, however many of them (maybe up to half - this should be quantified) do not. Therefore, the claim (L277-278) that "the pan-midgut driver were broadly distributed across the cell clusters" is misleading. Moreover, the hypothesis that "low expressing driver lines may result in particularly sparse labelling" (L331-333) is at least partially wrong, as Fig S13 shows that the same Gal4 driver can lead to very different levels of barcode coverage.

• Comparisons between TaG-EM and other, simpler methods for labelling individual cell populations are missing. For example, how would TaG-EM compare with expression of different fluorescent reporters, or a strategy based on the brainbow/flybow principle?

• FACS data is missing throughout the paper. The authors should include data from their comparative flow cytometry experiment of TaG-EM cells with or without additional hexameric GFP, as well as FSC/SSC and fluorescence scatter plots for the FACS steps that they performed prior to scRNA-seq, at least in supplementary figures.

• The authors should show the whole data described in L229, including the cluster that they chose to delete. At least, they should provide more information about how many cells were removed. In any case, the fact that their data still contains a large number of debris and dead cells despite sorting out PI negative cells with FACS and filtering low abundance barcodes with Cellranger is concerning.

Overall, although a method for genetic tagging cell populations prior to multiplexing in single-cell experiments would be extremely useful, the method presented here is inadequate. However, despite all the weaknesses listed above, the idea of barcodes expressed specifically in cells of interest deserves more consideration. If the authors manage to improve their design to resolve the major issues and demonstrate the benefits of their method more clearly, then TaG-EM could become an interesting option for certain applications.

Comments on revisions:

The authors have addressed many important points, providing reassurances about the initial weaknesses of their work. Although the TaG-EM is unlikely to have a significant influence on the field due to its limited benefits, the results are now sound and provide the reader with an unbiased view of the possibilities and limitations of the method.

---

## [Referee Report · Reviewer #2 (Public review)]

The authors developed the TaG-EM system to address challenges in multiplexing *Drosophila* samples for behavioral and transcriptomic studies. This system integrates DNA barcodes upstream of the polyadenylation site in a UAS-GFP construct, enabling pooled behavioral measurements and cell type tracking in scRNA-seq experiments. The revised manuscript expands on the utility of TaG-EM by demonstrating its application to complex assays, such as larval gut motility, and provides a refined analysis of its limitations and cost-effectiveness.

Strengths

(1) Novelty and Scope: The study demonstrates the potential for TaG-EM to streamline multiplexing in both behavioral and transcriptomic contexts. The additional application to labor-intensive larval gut motility assays highlights its scalability and practical utility.

(2) Data Quality and Clarity: Figures and supplemental data are mostly clear and significantly enhanced in the revised manuscript. The addition of Supplemental Figures 18-21 addresses initial concerns about scRNA-seq data and driver characterization.

(3) Cost-Effectiveness Analysis: New analyses of labor and cost savings (e.g., Supplemental Figure 8) provide a practical perspective.

(4) Improvements in Barcode Detection and Analysis: Enhanced enrichment protocols (Supplemental Figures 18-19) demonstrate progress in addressing limitations of barcode detection and increase the detection rate of labeled cells.

Weaknesses

(1) Barcode Detection Efficiency: While improvements are noted, the low barcode detection rate (~37% in optimized conditions) limits the method's scalability in some applications, such as single-cell sequencing experiments with complex cell populations.

(2) Sparse Labeling: Sparse labeling of cell populations, particularly in scRNA-seq assays, remains a concern. Variability in driver strength and regional expression introduces inconsistencies in labeling density.

(3) Behavioral Applications: The utility of TaG-EM in quantifying more complex behaviors remains underexplored, limiting the generalizability of the method beyond simpler assays like phototaxis and oviposition.

(4) Driver Line Characterization: While improvements in driver line characterization were made, variability in expression patterns and sparse labeling emphasize the need for further refinement of constructs and systematic backcrossing to standardize the genetic background.

---

## [Author Response]

The following is the authors’ response to the original reviews.

**Public Reviews:**

**Reviewer #1 (Public Review):**
The aim of this paper is to describe a novel method for genetic labelling of animals or cell populations, using a system of DNA/RNA barcodes.Strengths:• The author's attempt at providing a straightforward method for multiplexing *Drosophila* samples prior to scRNA-seq is commendable. The perspective of being able to load multiple samples on a 10X Chromium without antibody labelling is appealing.• The authors are generally honest about potential issues in their method, and areas that would benefit from future improvement.• The article reads well. Graphs and figures are clear and easy to understand.

We thank the reviewer for these positive comments.

Weaknesses:• The usefulness of TaG-EM for phototaxis, egg laying or fecundity experiments is questionable. The behaviours presented here are all easily quantifiable, either manually or using automated image-based quantification, even when they include a relatively large number of groups and replicates. Despite their claims (e.g., L311-313), the authors do not present any real evidence about the cost- or time-effectiveness of their method in comparison to existing quantification methods.

While the behaviors that were quantified in the original manuscript were indeed relatively easy to quantify through other methods, they nonetheless demonstrated that sequencing-based TaG-EM measurements faithfully recapitulated manual behavioral measurements. In response to the reviewer’s comment, we have added additional experiments that demonstrate the utility of TaG-EM-based behavioral quantification in the context of a more labor-intensive phenotypic assay (measuring gut motility via food transit times in *Drosophila* larvae, Figure 4, Supplemental Figure 7). We found that food transit times in the presence and absence of caffeine are subtly different and that, as with larger effect size behaviors, TaG-EM data recapitulates the results of the manual assay. This experiment demonstrates both that TaG-EM can be used to streamline labor-intensive behavioral assays (we have included an estimate of the savings in hands-on labor for this assay by using a multiplexed sequencing approach, Supplemental Figure 8) and that TaG-EM can quantify small differences between experimental groups. We also note in the discussion that an additional benefit of TaGEM-based behavioral assays is that the observed is blinded as to the experimental conditions as they are intermingled in a single multiplexed assay. We have added the following text to the paper describing these experiments.

Results:

“Quantifying food transit time in the larval gut using TaG-EM

Gut motility defects underlie a number of functional gastrointestinal disorders in humans (Keller et al., 2018). To study gut motility in *Drosophila*, we have developed an assay based on the time it takes a food bolus to transit the larval gut (Figure 4A), similar to approaches that have been employed for studying the role of the microbiome in human gut motility (Asnicar et al., 2021). Third instar larvae were starved for 90 minutes and then fed food containing a blue dye. After 60 minutes, larvae in which a blue bolus of food was visible were transferred to plates containing non-dyed food, and food transit (indicated by loss of the blue food bolus) was scored every 30 minutes for five hours (Supplemental Figure 7).

Because this assay is highly labor-intensive and requires hands-on effort for the entire five-hour observation period, there is a limit on how many conditions or replicates can be scored in one session (~8 plates maximum). Thus, we decided to test whether food transit could be quantified in a more streamlined and scalable fashion by using TaG-EM (Figure 4B). Using the manual assay, we observed that while caffeinecontaining food is aversive to larvae, the presence of caffeine reduces transit time through the gut (Figure 4C, Supplemental Figure 7). This is consistent with previous observations in adult flies that bitter compounds (including caffeine) activate enteric neurons via serotonin-mediated signaling and promote gut motility (Yao and Scott, 2022). We tested whether TaG-EM could be used to measure the effect of caffeine on food transit time in larvae. As with prior behavioral tests, the TaG-EM data recapitulated the results seen in the manual assay (Figure 4D). Conducting the transit assay via TaGEM enables several labor-saving steps. First, rather than counting the number of larvae with and without a food bolus at each time point, one simply needs to transfer nonbolus-containing larvae to a collection tube. Second, because the TaG-EM lines are genetically barcoded, all the conditions can be tested at once on a single plate, removing the need to separately count each replicate of each experimental condition. This reduces the hands-on time for the assay to just a few minutes per hour. A summary of the anticipated cost and labor savings for the TaG-EM-based food transit assay is shown in Supplemental Figure 8.”

Discussion:

“While the utility of TaG-EM barcode-based quantification will vary based on the number of conditions being analyzed and the ease of quantifying the behavior or phenotype by other means, we demonstrate that TaG-EM can be employed to cost-effectively streamline labor-intensive assays and to quantify phenotypes with small effect sizes (Figure 4, Supplemental Figure 8). An additional benefit of multiplexed TaG-EM behavioral measurements is that the experimental conditions are effectively blinded as the multiplexed conditions are intermingled in a single assay.”

Methods:

“Larval gut motility experiments

Preparing Yeast Food Plates

Yeast agar plates were prepared by making a solution containing 20% Red Star Active Dry Yeast 32oz (Red Star Yeast) and 2.4% Agar Powder/Flakes (Fisher) and a separate solution containing 20% Glucose (Sigma-Aldrich). Both mixtures were autoclaved with a 45-minute liquid cycle and then transferred to a water bath at 55ºC. After cooling to 55ºC, the solutions were combined and mixed, and approximately 5 mL of the combined solution was transferred into 100 x 15 mm petri dishes (VWR) in a PCR hood or contamination-free area. For blue-dyed yeast food plates, 0.4% Blue Food Color (McCormick) was added to the yeast solution. For the caffeine assays, 300 µL of a solution of 100 mM 99% pure caffeine (Sigma-Aldrich) was pipetted onto the blue-dyed yeast plate and allowed to absorb into the food during the 90-minute starvation period.

Manual Gut Motility Assay

Third instar *Drosophila* larvae were transferred to empty conical tubes that had been misted with water to prevent the larvae from drying out. After a 90-minute starvation period the larvae were moved from the conical to a blue-dyed yeast plate with or without caffeine and allowed to feed for 60 minutes. Following the feeding period, the larvae were transferred to an undyed yeast plate. Larvae were scored for the presence or absence of a food bolus every 30 minutes over a 5-hour period. Up to 8 experimental replicates/conditions were scored simultaneously.

TaG-EM Gut Motility Assay

Third instar larvae were starved and fed blue dye-containing food with or without caffeine as described above. An equal number of larvae from each experimental condition/replicate were transferred to an undyed yeast plate. During the 5-hour observation period, larvae were examined every 30 minutes and larvae lacking a food bolus were transferred to a microcentrifuge tube labeled for the timepoint. Any larvae that died during the experiment were placed in a separate microcentrifuge tube and any larvae that failed to pass the food bolus were transferred to a microcentrifuge tube at the end of the experiment. DNA was extracted from the larvae in each tube and TaG-EM barcode libraries were prepared and sequenced as described above.”

• Behavioural assays presented in this article have clear outcomes, with large effect sizes, and therefore do not really challenge the efficiency of TaG-EM. By showing a Tmaze in Fig 1B, the authors suggest that their method could be used to quantify more complex behaviours. Not exploring this possibility in this manuscript seems like a missed opportunity.

See the response to the previous point.

• Experiments in Figs S3 and S6 suggest that some tags have a detrimental effect on certain behaviours or on GFP expression. Whereas the authors rightly acknowledge these issues, they do not investigate their causes. Unfortunately, this question the overall suitability of TaG-EM, as other barcodes may also affect certain aspects of the animal's physiology or behaviour. Revising barcode design will be crucial to make sure that sequences with potential regulatory function are excluded.

We have determined that the barcode (BC#8) that had no detectable Gal4induced gene expression in Figure S6 (now Supplemental Figure 9) has a deletion in the GFP coding region that ablates GFP function. Interestingly, the expressed TaG-EM barcode transcript is still detectable in single cell sequencing experiments, but obviously this line cannot be used for cell enrichment (at least based solely on GFP expression from the TaG-EM construct). While it is unclear how this line came to have a lesion in the GFP gene, we have subsequently generated >150 additional TaG-EM stocks and we have tested the GFP expression of these newly established stocks by crossing them to *Mhc-Gal4*. All of the additional stocks had GFP expression in the expected pattern, indicating that the BC#8 construct is an outlier with respect to inducibility of GFP. We have added the following text to the results section to address this point:

“No GFP expression was visible for TaG-EM barcode number 8, which upon molecular characterization had an 853 bp deletion within the GFP coding region (data not shown). We generated and tested GFP expression of an additional 156 TaG-EM barcode lines (Alegria et al., 2024), by crossing them to Mhc-Gal4 and observing expression in the adult thorax. All 156 additional TaG-EM lines had robust GFP expression (data not shown).”

It is certainly the case that future improvements to the construct design may be necessary or desirable and that back-crossing could likely be used to alleviate line-toline differences for specific phenotypes, we also address this point in the discussion with the following text:

“We excluded this poor performing barcode line from the fecundity tests, however, backcrossing is often used to bring reagents into a consistent genetic background for behavioral experiments and could also potentially be used to address behavior-specific issues with specific TaG-EM lines. In addition, other strategies such as averaging across multiple barcode lines or permutation of barcode assignment across replicates could also mitigate such deficiencies.”

• For their single-cell experiments, the authors have used the 10X Genomics method, which relies on sequencing just a short segment of each transcript (usually 50-250bp - unknown for this study as read length information was not provided) to enable its identification, with the matching paired-end read providing cell barcode and UMI information (Macosko et al., 2015). With average fragment length after tagmentation usually ranging from 300-700bp, a large number of GFP reads will likely not include the 14bp TaG-EM barcode.

The 10x Genomics 3’ workflows that were used for sequencing TaG-EM samples reads the cell barcode and UMI in read one and the expressed RNA sequence in read two. We sequenced the samples shown in Figure 5 in the initial manuscript using a run configuration that generated 150 bp for read two. The TaG-EM barcodes are located just upstream of the poly-adenylation sites (based on the sequencing data, we observe two different poly-A sites and the TaG-EM barcode is located 35 and 60 bp upstream of these sites). Based on the location of the TaG-EM barcodes,150 bp reads is sufficient to see the barcode in any GFP-associated read (when using the 3’ gene expression workflow). In addition to detecting the expression of the TaG-EM barcodes in the 10x Genomics gene expression library, it is possible to make a separate library that enriches the barcode sequence (similar to hashtag or CITE-Seq feature barcode libraries). We have added experimental data where we successfully performed an enrichment of the TaG-EM barcodes and sequenced this as a separate hashtag library (Supplemental Figure 18). We have added text to the results describing this work and also included a detailed information in the methods for performing TaG-EM barcode enrichment during 10x library prep.

Results:

“In antibody-conjugated oligo cell hashing approaches, sparsity of barcode representation is overcome by spiking in an additional primer at the cDNA amplification step and amplifying the hashtag oligo by PCR. We employed a similar approach to attempt to enrich for TaG-EM barcodes in an additional library sequenced separately from the 10x Genomics gene expression library. Our initial attempts at barcode enrichment using spike-in and enrichment primers corresponding to the TaG-EM PCR handle were unsuccessful (Supplemental Figure 18). However, we subsequently optimized the TaG-EM barcode enrichment by (1) using a longer spike-in primer that more closely matches the annealing temperature used during the 10x Genomics cDNA creation step, and (2) using a nested PCR approach to amplify the cell-barcode and unique molecular identifier (UMI)-labeled TaG-EM barcodes (Supplemental Figure 18). Using the enriched library, TaG-EM barcodes were detected in nearly 100% of the cells at high sequencing depths (Supplemental Figure 19). However, although we used a polymerase that has been engineered to have high processivity and that has been shown to reduce the formation of chimeric reads in other contexts (Gohl et al., 2016), it is possible that PCR chimeras could lead to unreliable detection events for some cells. Indeed, many cells had a mixture of barcodes detected with low counts and single or low numbers of associated UMIs. To assess the reliability of detection, we analyzed the correlation between barcodes detected in the gene expression library and the enriched TaG-EM barcode library as a function of the purity of TaG-EM barcode detection for each cell (the percentage of the most abundant detected TaG-EM barcode, Supplemental Figure 19). For TaG-EM barcode detections where the most abundance barcode was a high percentage of the total barcode reads detected (~75%-99.99%), there was a high correlation between the barcode detected in the gene expression library and the enriched TaG-EM barcode library. Below this threshold, the correlation was substantially reduced.

In the enriched library, we identified 26.8% of cells with a TaG-EM barcode reliably detected, a very modest improvement over the gene expression library alone (23.96%), indicating that at least for this experiment, the main constraint is sufficient expression of the TaG-EM barcode and not detection. To identify TaG-EM barcodes in the combined dataset, we counted a positive detection as any barcode either identified in the gene expression library or any barcode identified in the enriched library with a purity of >75%. In the case of conflicting barcode calls, we assigned the barcode that was detected directly in the gene expression library. This increased the total fraction of cells where a barcode was identified to approximately 37% (Figure 6B).”

Methods:

“The resulting pool was prepared for sequencing following the 10x Genomics Single Cell 3’ protocol (version CG000315 Rev C), At step 2.2 of the protocol, cDNA amplification, 1 µl of TaG-EM spike-in primer (10 µM) was added to the reaction to amplify cDNA with the TaG-EM barcode. Gene expression cDNA and TaG-EM cDNA were separated using a double-sided SPRIselect (Beckman Coulter) bead clean up following 10x Genomics Single Cell 3’ Feature Barcode protocol, step 2.3 (version CG000317 Rev E). The gene expression cDNA was created into a library following the CG000315 Rev C protocol starting at section 3. Custom nested primers were used for enrichment of TaG-EM barcodes after cDNA creation using PCR. The following primers were tested (see Supplemental Figure 18):

UMGC_IL_TaGEM_SpikeIn_v1:

GTGACTGGAGTTCAGACGTGTGCTCTTCCGATCTCTTCCAACAACCGGAAGT*G*A UMGC_IL_TaGEM_SpikeIn_v2:

GTGACTGGAGTTCAGACGTGTGCTCTTCCGATCTGCAGCTTATAACTTCCAACAACCGGAAGT*G*A

UMGC_IL_TaGEM_SpikeIn_v3:

TGTGCTCTTCCGATCTGCAGCTTATAACTTCCAACAACCGGAAGT*G*A D701_TaGEM:

CAAGCAGAAGACGGCATACGAGATCGAGTAATGTGACTGGAGTTCAGACGTGTGCTCTTCCGATCTGCAGC*T*T

SI PCR Primer:

AATGATACGGCGACCACCGAGATCTACACTCTTTCCCTACACGACGC*T*C

UMGC_IL_DoubleNest:

GTGACTGGAGTTCAGACGTGTGCTCTTCCGATCTGCAGCTTATAACTTCCAACAACCGG*A*A

P5: AATGATACGGCGACCACCGA

D701:

GATCGGAAGAGCACACGTCTGAACTCCAGTCACATTACTCGATCTCGTATGCCGTCTTCTGCTTG

D702:

GATCGGAAGAGCACACGTCTGAACTCCAGTCACTCCGGAGAATCTCGTATGCCGTCTTCTGCTTG

After multiple optimization trials, the following steps yielded ~96% on-target reads for the TaG-EM library (Supplemental Figure 18, note that for the enriched barcode data shown in Figure 6 and Supplemental Figure 19, a similar amplification protocol was used TaG-EM barcodes were amplified from the gene expression library cDNA and not the SPRI-selected barcode pool). TaG-EM cDNA was amplified with the following PCR reaction: 5 µl purified TaG-EM cDNA, 50 µl 2x KAPA HiFi ReadyMix (Roche), 2.5 µl UMGC_IL_DoubleNest primer (10 µM), 2.5 µl SI_PCR primer (10 µM), and 40 µl nuclease-free water. The reaction was amplified using the following cycling conditions: 98ºC for 2 minutes, followed by 15 cycles of 98ºC for 20 seconds, 63ºC for 30 seconds, 72ºC for 20 seconds, followed by 72ºC for 5 minutes. After the first PCR, the amplified cDNA was purified with a 1.2x SPRIselect (Beckman Coulter) bead cleanup with 80% ethanol washes and eluted into 40 µL of nuclease-water. A second round of PCR was run with following reaction: 5 µl purified TaG-EM cDNA, 50 µl 2x KAPA HiFi ReadyMix (Roche), 2.5 µl D702 primer (10 µM), 2.5 µl p5 Primer (10 µM), and 40 µl nuclease-free water. The reaction was amplified using the following cycling conditions: 98ºC for 2 minutes, followed by 10 cycles of 98ºC for 20 seconds, 63ºC for 30 seconds, 72ºC for 20 seconds, followed by 72ºC for 5 minutes. After the second PCR, the amplified cDNA was purified with a 1.2x SPRIselect (Beckman Coulter) bead cleanup with 80% ethanol washes and eluted into 40uL of nuclease-water. The resulting 3’ gene expression library and TaG-EM enrichment library were sequenced together following Scenario 1 of the BioLegend “Total-Seq-A Antibodies and Cell Hashing with 10x Single Cell 3’ Reagents Kit v3 or v3.1” protocol. Additional sequencing of the enriched TaG-EM library also done following Scenario 2 from the same protocol.”

When a given cell barcode is not associated with any TaG-EM barcode, then demultiplexing is impossible. This is a major problem, which is particularly visible in Figs 5 and S13. In 5F, BC4 is only detected in a couple of dozen cells, even though the Jon99Ciii marker of enterocytes is present in a much larger population (Fig 5C). Therefore, in this particular case, TaG-EM fails to detect most of the GFP-expressing cells.

Figure 5 in the original manuscript represented data from an experiment in which there were eight different TaG-EM barcoded samples present, including four replicates of the pan-midgut driver (each of which included enterocyte populations). One would not expect the BC4 enterocyte driver expression to be observed in all of the Jon99Ciii cells, since the majority of the GFP+ cells shown in the UMAP plot were likely derived from and are labeled by the pan-midgut driver-associated barcodes. Thus, the design and presentation of this particular experiment (in particular, the presence of eight distinct samples in the dataset) is making the detection of the TaG-EM barcodes look sparser than it actually is. We have added a panel in both Figure 6B and Supplemental Figure 17B that shows the overall detection of barcodes in the enriched barcode library and gene expression library or the gene expression library only, respectively, for this experiment.

However, the reviewer’s overall point regarding barcode detection is still valid in that if we consider all eight barcodes, we only see TaG-EM barcode labeling associated with about a quarter of all the cells in this gene expression library, or about 37% of cells when we include the enriched TaG-EM barcode library. While improving barcode detection will improve the yield and is necessary for some applications (such as robust detection of multiplets), we would argue that even at the current level of success this approach has significant utility. First, if one’s goal is to unambiguously label a cell cluster and trace it to a defined cell population in vivo, sparse labeling may be sufficient. Second, demultiplexing is still possible (as we demonstrate) but involves a trade off in yield (not every cell is recovered and there is some extra sequencing cost as some sequenced cells cannot be assigned to a barcode).

Similarly, in S13, most cells should express one of the four barcodes, however many of them (maybe up to half - this should be quantified) do not. Therefore, the claim (L277278) that "the pan-midgut driver were broadly distributed across the cell clusters" is misleading. Moreover, the hypothesis that "low expressing driver lines may result in particularly sparse labelling" (L331-333) is at least partially wrong, as Fig S13 shows that the same Gal4 driver can lead to very different levels of barcode coverage.

As described above, since this experiment included eight different TaG-EM barcodes expressed by five different drivers, the expectation is that only about half of the cells in Figure S13 (now Figure S20) should express a TaG-EM barcode. It is not clear why BC2 is underrepresented in terms of the number of cells labeled and BC7 is overrepresented. We agree with the reviewer that this should be described more accurately in the paper and that it does impact our interpretation related to driver strength and barcode detection. We have revised this sentence in the discussion and also added additional text in the results describing the within driver variability seen in this experiment.

Results text:

“As expected, the barcodes expressed by the pan-midgut driver were broadly distributed across the cell clusters (Supplemental Figure 20). However, the number of cells recovered varied significantly among the four pan-midgut driver associated barcodes.”

Discussion text:

“It is likely that the strength of the Gal4 driver contributes to the labeling density. However, we also observed variable recovery of TaG-EM barcodes that were all driven by the same pan-midgut Gal4 driver (Supplemental Figure 20).”

• Comparisons between TaG-EM and other, simpler methods for labelling individual cell populations are missing. For example, how would TaG-EM compare with expression of different fluorescent reporters, or a strategy based on the brainbow/flybow principle?

The advantage of TaG-EM is that an arbitrarily large number of DNA barcodes can be used (contingent upon the availability of transgenic lines – we described 20 barcoded lines in our initial manuscript and we have now extended this collection to over 170 lines), while the number of distinguishable FPs is much lower. Brainbow/Flybow uses combinatorial expression of different FPs, but because this combinatorial expression is stochastic, tracing a single cell transcriptome to a defined cell population in vivo based on the FP signature of a Brainbow animal would likely not be possible (and would almost certainly be impossible at scale).

• FACS data is missing throughout the paper. The authors should include data from their comparative flow cytometry experiment of TaG-EM cells with or without additional hexameric GFP, as well as FSC/SSC and fluorescence scatter plots for the FACS steps that they performed prior to scRNA-seq, at least in supplementary figures.

We have added Supplemental Figures with the FACS data for all of the single cell sequencing data presented in the manuscript (Supplemental Figures 12 and 14).

• The authors should show the whole data described in L229, including the cluster that they chose to delete. At least, they should provide more information about how many cells were removed. In any case, the fact that their data still contains a large number of debris and dead cells despite sorting out PI negative cells with FACS and filtering low abundance barcodes with Cellranger is concerning.

This description was referring to the unprocessed Cellranger output (not filtered for low abundance barcodes). Prior to filtering for cell barcodes with high mitochondria or rRNA (or other processing in Seurat/Scanpy), we saw two clusters, one with low UMI counts and enrichment of mitochondrial genes (see Cellranger report below).

These cell barcodes were removed by downstream quality filtering and the remaining cells showed expression of expected intestinal stem cell and enteroblast marker genes.

Overall, although a method for genetic tagging cell populations prior to multiplexing in single-cell experiments would be extremely useful, the method presented here is inadequate. However, despite all the weaknesses listed above, the idea of barcodes expressed specifically in cells of interest deserves more consideration. If the authors manage to improve their design to resolve the major issues and demonstrate the benefits of their method more clearly, then TaG-EM could become an interesting option for certain applications.

We thank the reviewer for this comment and hope that the above responses and additional experiments and data that we have added have helped to alleviate the noted weaknesses.

**Reviewer #2 (Public Review):**
In this manuscript, Mendana et al developed a multiplexing method - Targeted Genetically-Encoded Multiplexing or TaG-EM - by inserting a DNA barcode upstream of the polyadenylation site in a Gal4-inducible UAS-GFP construct. This Multiplexing method can be used for population-scale behavioral measurements or can potentially be used in single-cell sequencing experiments to pool flies from different populations. The authors created 20 distinctly barcoded fly lines. First, TaG-EM was used to measure phototaxis and oviposition behaviors. Then, TaG-EM was applied to the fly gut cell types to demonstrate its applications in single-cell RNA-seq for cell type annotation and cell origin retrieving.This TaG-EM system can be useful for multiplexed behavioral studies from nextgeneration sequencing (NGS) of pooled samples and for Transcriptomic Studies. I don't have major concerns for the first application, but I think the scRNA-seq part has several major issues and needs to be further optimized.Major concerns:(1) It seems the barcode detection rate is low according to Fig S9 and Fig 5F, J and N. Could the authors evaluate the detection rate? If the detection rate is too low, it can cause problems when it is used to decode cell types.

See responses to Reviewer #1 on this topic above.

(2) Unsuccessful amplification of TaG-EM barcodes: The authors attempted to amplify the TaG-EM barcodes in parallel to the gene expression library preparation but encountered difficulties, as the resulting sequencing reads were predominantly offtarget. This unsuccessful amplification raises concerns about the reliability and feasibility of this amplification approach, which could affect the detection and analysis of the TaG-EM barcodes in future experiments.

As noted above, we have now established a successful amplification protocol for the TaG-EM barcodes. This data is shown in Figure 6, and Supplemental Figures 18-19 and we have included a detailed information in the methods for performing TaG-EM barcode enrichment during 10x library prep. We have also included code in the paper’s Github repository for assigning TaG-EM barcodes from the enriched library to the associated 10x Genomics cell barcodes.

(3) For Fig 5, the singe-cell clusters are not annotated. It is not clear what cell types are corresponding to which clusters. So, it is difficult to evaluate the accuracy of the assignment of barcodes.

We have added annotation information for the cell clusters based on expression of cell-type-specific marker genes (Figure 6A, Supplemental Figures 16-17).

(4) The scRNA-seq UMAP in Fig 5 is a bit strange to me. The fly gut epithelium contains only a few major cell types, including ISC, EB, EC, and EE. However, the authors showed 38 clusters in fig 5B. It is true that some cell types, like EE (Guo et al., 2019, Cell Reports), have sub-populations, but I don't expect they will form these many subtypes. There are many peripheral small clusters that are not shown in other gut scRNAseq studies (Hung et al., 2020; Li et al., 2022 Fly Cell Atlas; Lu et al., 2023 Aging Fly Cell Atlas). I suggest the authors try different data-processing methods to validate their clustering result.

For all of the single cell experiments, after doublet and ambient RNA removal (as suggested below), we have reclustered the datasets and evaluated different resolutions using Clustree. As the Reviewer points out, there are different EE subtypes, as well as regionalized expression differences in EC and other cell populations, so more than four clusters are expected (an analysis of the adult midgut identified 22 distinct cell types). With this revised analysis our results more closely match the cell populations observed in other studies (though it should be noted that the referenced studies largely focus on the adult and not the larval stage).

(5) Different gut drivers, PMC-, PC-, EB-, EC-, and EE-GAL4, were used. The authors should carefully characterize these GAL4 expression in larval guts and validate sequencing data. For example, does the ratio of each cell type in Fig 5B reflect the in vivo cell type ratio? The authors used cell-type markers mostly based on the knowledge from adult guts, but there are significant morphological and cell ratio differences between larval and adult guts (e.g., Mathur...Ohlstein, 2010 Science).

We have characterized the PC driver which is highlighted in Supplemental Figure 13, and the EC and EE drivers which are highlighted in Figure 6G-N in detail in larval guts and have added this data to the paper (Supplemental Figure 21). The EB driver was not characterized histologically as EB-specific antibodies are not currently available. The PMG-Gal4 line exhibits strong expression throughout the larval gut (Figure 5B) and barcodes are recovered from essentially all of the larval gut cell clusters using this driver (Supplemental Figure 20). We don’t necessarily expect the ratios of cells observed in the scRNA-Seq data to reflect the ratios typically observed in the gut as we performed pooled flow sorting on a multiplexed set of eight genotypes and driver expression levels, flow sorting, and possibly other processing steps could all influence the relative abundance of different cell types. However, detailed characterization of these driver lines did reveal spatial expression patterns that help explain aspects of the scRNA-Seq data. We have also added the following text to the paper to further describe the characterization of the drivers:

Results:

“Detailed characterization of the EC-Gal4 line indicated that although this line labeled a high percentage of enterocytes, expression was restricted to an area at the anterior and middle of the midgut, with gaps between these regions and at the posterior (Supplemental Figure 21). This could explain the absence of subsets of enterocytes, such as those labeled by *betaTry*, which exhibits regional expression in R2 of the adult midgut (Buchon et al., 2013).”

“Detailed characterization of the EE-Gal4 driver line indicated that ~80-85% of Prospero-positive enteroendocrine cells are labeled in the anterior and middle of the larval midgut, with a lower percentage (~65%) of Prospero-positive cells labeled in the posterior midgut (Supplemental Figure 21). As with the enterocyte labeling, and consistent with the Gal4 driver expression pattern, the EE-Gal4 expressed TaG-EM barcode 9 did not label all classes of enteroendocrine cells and other clusters of presumptive enteroendocrine cells expressing other neuropeptides such as *Orcokinin*, *AstA*, and *AstC*, or neuropeptide receptors such as *CCHa2* (not shown) were also observed.”

Methods:

“Dissection and immunostaining

Midguts from third instar larvae of driver lines crossed to UAS-GFP.nls or UAS-mCherry were dissected in 1xPBS and fixed with 4% paraformaldehyde (PFA) overnight at 4ºC. Fixed samples were washed with 0.1% PBTx (1xPBS + 0.1% Triton X-100) three times for 10 minutes each and blocked in PBTxGS (0.1% PBTx + 3% Normal Goat Serum) for 2–4 hours at RT. After blocking, midguts were incubated in primary antibody solution overnight at 4ºC. The next day samples were washed with 0.1% PBTx three times for 20 minutes each and were incubated in secondary antibody solution for 2–3 hours at RT (protected from light) followed by three washes with 0.1% PBTx for 20 minutes each. One µg/ml DAPI solution prepared in 0.1% PBTx was added to the sample and incubated for 10 minutes followed by washing with 0.1% PBTx three times for 10 minutes each. Finally, samples were mounted on a slide glass with 70% glycerol and imaged using a Nikon AX R confocal microscope. Confocal images were processed using Fiji software.

The primary antibodies used were rabbit anti-GFP (A6455,1:1000 Invitrogen), mouse anti-mCherry (3A11, 1:20 DSHB), mouse anti-Prospero (MR1A, 1:50 DSHB) and mouse anti-Pdm1 (Nub 2D4, 1:30 DSHB). The secondary antibodies used were goat antimouse and goat anti-rabbit IgG conjugated to Alexa 647 and Alexa 488 (1:200) (Invitrogen), respectively. Five larval gut specimens per Gal4 line were dissected and examined.”

(6) Doublets are removed based on the co-expression of two barcodes in Fig 5A. However, there are also other possible doublets, for example, from the same barcode cells or when one cell doesn't have detectable barcode. Did the authors try other computational approaches to remove doublets, like DoubleFinder (McGinnis et al., 2019) and Scrublet (Wolock et al., 2019)?

We have included DoubleFinder-based doublet removal in our data analysis pipeline. This is now described in the methods (see below).

(7) Did the authors remove ambient RNA which is a common issue for scRNA-seq experiments?

We have also used DecontX to remove ambient RNA. This is now described in the methods:

“Datasets were first mapped and analyzed using the Cell Ranger analysis pipeline (10x Genomics). A custom *Drosophila* genome reference was made by combining the BDGP.28 reference genome assembly and Ensembl gene annotations. Custom gene definitions for each of the TaG-EM barcodes were added to the fasta genome file and .gtf gene annotation file. A Cell Ranger reference package was generated with the Cell Ranger *mkref* command. Subsequent single-cell data analysis was performed using the R package Seurat (Satija et al., 2015). Cells expressing less than 200 genes and genes expressed in fewer than three cells were filtered from the expression matrix. Next, percent mitochondrial reads, percent ribosomal reads cells counts, and cell features were graphed to determine optimal filtering parameters. DecontX (Yang et al., 2020) was used to identify empty droplets, to evaluate ambient RNA contamination, and to remove empty cells and cells with high ambient RNA expression. DoubletFinder (McGinnis et al., 2019) to identify droplet multiplets and remove cells classified as multiplets. Clustree (Zappia and Oshlack, 2018) was used to visualize different clustering resolutions and to determine the optimal clustering resolution for downstream analysis. Finally, SingleR (Aran et al., 2019) was used for automated cell annotation with a gut single-cell reference from the Fly Cell Atlas (Li et al., 2022). The dataset was manually annotated using the expression patterns of marker genes known to be associated with cell types of interest. To correlate TaG-EM barcodes with cell IDs in the enriched TaG-EM barcode library, a custom Python script was used (TaGEM_barcode_Cell_barcode_correlation.py), which is available via Github: https://github.com/darylgohl/TaG-EM.”

(8) Why does TaG-EM barcode #4, driven by EC-GAL4, not label other classes of enterocyte cells such as betaTry+ positive ECs (Figures 5D-E)? similarly, why does TaG-EM barcode #9, driven by EE-GAL4, not label all EEs? Again, it is difficult to evaluate this part without proper data processing and accurate cell type annotation.

As noted in the response to a comment by Reviewer #1 above, part of this apparent sparsity of labeling is due to the way that this experiment was designed and visualized. We have added a new Figure panel in both Figure 6B and Supplemental Figure 17B that shows the overall detection of barcodes in the enriched barcode library and gene expression library or the gene expression library only, respectively, to better illustrate the efficacy of barcode detection. See also the response to point 5 above. Both the lack of labelling of betaTry+ ECs and subsets of EEs is consistent with the expression patterns of the EC-Gal4 and EE-Gal4 drivers.

(9) For Figure 2, when the authors tested different combinations of groups with various numbers of barcodes. They found remarkable consistency for the even groups. Once the numbers start to increase to 64, barcode abundance becomes highly variable (range of 12-18% for both male and female). I think this would be problematic because the differences seen in two groups for example may be due to the barcode selection rather than an actual biologically meaningful difference.

While there is some barcode-to-barcode variability for different amplification conditions, the magnitude of this variation is relatively consistent across the conditions tested. We looked at the coefficient of variation for the evenly pooled barcodes or for the staggered barcodes pooled at different relative levels. While the absolute magnitude of the variation is higher for the highly abundant barcodes in the staggered conditions, the CVs for these conditions (0.186 for female flies and for 0.163 male flies) were only slightly above the mean CV (0.125) for all conditions (see Supplemental Figure 3):

We have added this analysis as Supplemental Figure 3 and added the following text to the paper:

“The coefficients of variation were largely consistent for groups of TaG-EM barcodes pooled evenly or at different levels within the staggered pools (Supplemental Figure 3).”

(10) Barcode #14 cannot be reliably detected in oviposition experiment. This suggests that the BC 14 fly line might have additional mutations in the attp2 chromosome arm that affects this behavior. Perhaps other barcode lines also have unknown mutations and would cause issues for other untested behaviors. One possible solution is to backcross all 20 lines with the same genetic background wild-type flies for >7 generations to make all these lines to have the same (or very similar) genetic background. This strategy is common for aging and behavior assays.

See response to Reviewer #1 above on this topic.

**Reviewer #3 (Public Review):**
The work addresses challenges in linking anatomical information to transcriptomic data in single-cell sequencing. It proposes a method called Targeted Genetically-Encoded Multiplexing (TaG-EM), which uses genetic barcoding in *Drosophila* to label specific cell populations in vivo. By inserting a DNA barcode near the polyadenylation site in a UASGFP construct, cells of interest can be identified during single-cell sequencing. TaG-EM enables various applications, including cell type identification, multiplet droplet detection, and barcoding experimental parameters. The study demonstrates that TaGEM barcodes can be decoded using next-generation sequencing for large-scale behavioral measurements. Overall, the results are solid in supporting the claims and will be useful for a broader fly community. I have only a few comments below:

We thank the reviewer for these positive comments.

Specific comments:(1) The authors mentioned that the results of structure pool tests in Fig. 2 showed a high level of quantitative accuracy in detecting the TaG-EM barcode abundance. Although the data were generally consistent with the input values in most cases, there were some obvious exceptions such as barcode 1 (under-represented) and barcodes 15, 20 (overrepresented). It would be great if the authors could comment on these and provide a guideline for choosing the appropriate barcode lines when implementing this TaG-EM method.

See the response to point 9 from Reviewer 2. Although there seem to be some systematic differences in barcode amplification, the coefficient of variation was relatively consistent across all of the barcode combinations and relative input levels that we examined. Our recommendation (described in the text) is to average across 3-4 independent barcodes (which yielded a R2 values of >0.99 with expected abundance in the structured pooled tests).

(2) In Supplemental Figure 6, the authors showed GFP antibody staining data with 20 different TaG-EM barcode lines. The variability in GFP antibody staining results among these different TaG-EM barcode lines concerns the use of these TaG-EM barcode lines for sequencing followed by FACS sorting of native GFP. I expected the native GFP expression would be weaker and much more variable than the GFP antibody staining results shown in Supplemental Figure 6. If this is the case, variation of tissue-specific expression of TaG-EM barcode lines will likely be a confounding factor.

Aside from barcode 8, which had a mutation in the GFP coding sequence, we did not see significant variability in expression levels either in the wing disc. Subtle differences seen in this figure most likely result from differences in larval staging. Similar consistent native (unstained) GFP expression of the TaG-EM constructs was seen in crosses with Mhc-Gal4 (described above).

(3) As the authors mentioned in the manuscript, multiple barcodes for one experimental condition would be a better experimental design. Could the authors suggest a recommended number of barcodes for each experiential condition? 3? 4? Or more?

See response to Reviewer #3, point number 1 above.

(3b) Also, it would be great if the authors could provide a short discussion on the cost of such TaG-EM method. For example, for the phototaxis assay, if it is much more expensive to perform TaG-EM as compared to manually scoring the preference index by videotaping, what would be the practical considerations or benefits of doing TaG-EM over manual scoring?

While this will vary depending on the assay and the scale at which one is conducting experiments, we have added an analysis of labor savings for the larval gut motility assay (Supplemental Figure 8). We have also added the following text to the Discussion describing some of the trade-offs to consider in assessing the potential benefit of incorporating TaG-EM into behavioral measurements:

“While the utility of TaG-EM barcode-based quantification will vary based on the number of conditions being analyzed and the ease of quantifying the behavior or phenotype by other means, we demonstrate that TaG-EM can be employed to cost-effectively streamline labor-intensive assays and to quantify phenotypes with small effect sizes (Figure 4, Supplemental Figure 8).”

**Recommendations for the authors:**
While recognising the potential of the TaG-EM methodology, we had a few major concerns that the authors might want to consider addressing:

As stated above, we are grateful to the reviewers and editor for their thoughtful comments. We have addressed many of the points below in our responses above, so we will briefly respond to these points and where relevant direct the reader to comments above.

(1) We were concerned about the efficacy of TaG-EM in assessing more complex behaviours than oviposition and phototaxis. We note that Barcode #14 cannot be reliably detected in oviposition experiment. This suggests that the BC 14 fly line might have additional mutations in the attp2 chromosome arm that affects this behavior. Perhaps other barcode lines also have unknown mutations and would cause issues for other untested behaviors. One possible solution is to back-cross all 20 lines with the same genetic background wild-type flies for >7 generations to make all these lines to have the same (or very similar) genetic background. This strategy is common for aging and behavior assays.

See response to Reviewer #1 and Reviewer #2, item 10, above.

(2) We were unable to assess the drop-out rates of the TaG-EM barcode from the sequencing. The barcode detection rate is low (Fig S9 and Fig 5F, J and N). This would be a considerable drawback (relating to both experimental design and cost), if a large proportion of the cells could not be assigned an identity.

See comments above addressing this point.

(3) The effectiveness of TaG-EM scRNA-seq on the larvae gut is not very effective - the cells are not well annotated, the barcodes seem not to have labelled expected cell types (ECs and EEs), and there is no validation of the Gal4 drivers in vivo.

See previous comments. We have addressed specific comments above on data processing and annotation, included a visualization of the overall effectiveness of labeling, added a protocol and data on enriched TaG-EM barcode libraries, and have added detailed characterization of the Gal4 drivers in the larval gut (Figure 6, Supplemental Figures 17-21).

(4) A formal assessment of the cost-effectiveness would be an important consideration in broad uptake of the methodology.

While this is difficult to do in a comprehensive manner given the breadth of potential applications, we have included estimates of labor savings for one of the behavioral assays that we tested (Supplemental Figure 8). We have also included a discussion of some of the factors that would make TaG-EM useful or cost-effective to apply for behavioral assays (see response to Reviewer #3, comment 3b, above). We have also added the following text to the discussion to address the cost considerations in applying TaG-EM for scRNA-Seq:

“For single cell RNA-Seq experiments, the cost savings of multiplexing is roughly the cost of a run divided by the number of independent lines multiplexed, plus labor savings by also being able to multiplex upstream flow cytometry, minus loss of unbarcoded cells. Our experiments indicated that for the specific drivers we tested TaG-EM barcodes are detected in around one quarter of the cells if relying on endogenous expression in the gene expression library, though this fraction was higher (~37%) if sequencing an enriched TaG-EM barcode library in parallel (Figure 6, Supplemental Figures 18-19).”

(5) Similarly, a formal assessment of the effect of the insertion on the variability in GFP expression and the behaviour needs to be documented.

See responses to Reviewer #1, Reviewer #2, item 9, and Reviewer #3, item 2 above.

**Reviewer #1 (Recommendations For The Authors):**
(in no particular order of importance)• L84-85: the authors should either expand, or remove this statement. Indeed, lack of replicates is only true if one ignores that each cell in an atlas is indeed a replicate. Therefore, depending on the approach or question, this statement is inaccurate.

This sentence was meant to refer to experiments where different experimental conditions are being compared and not to more descriptive studies such as cell atlases. We have revised this sentence to clarify.

“Outside of descriptive studies, these costs are also a barrier to including replicates to assess biological variability; consequently, a lack of biological replicates derived from independent samples is a common shortcoming of single-cell sequencing experiments.”

• L103-104: this sentence is unclear.

We have revised this sentence as follows:

“Genetically barcoded fly lines can also be used to enable highly multiplexed behavioral assays which can be read out using high throughput sequencing.”

• In Fig S1 it is unclear why there are more than 20 different sequences in panel B where the text and panel A only mention the generation of 20 distinct constructs. This should be better explained.

The following text was added to the Figure legend to explain this discrepancy:

“Because the TaG-EM barcode constructs were injected as a pool of 29 purified plasmids, some of the transgenic lines had inserts of the same construct. In total 20 unique lines were recovered from this round of injection.”

• It would be interesting to compare the efficiency of TaG-EM driven doublet removal (Fig 5A) with standard doublet-removing software (e.g., DoubletFinder, McGinnis et al., 2019).

We have done this comparison, which is now shown in Supplemental Figure 15.

• I would encourage the authors to check whether barcode representation in Fig S13 can be correlated to average library size, as one would expect libraries with shorter reads to be more likely to include the 14-bp barcode and therefore more accurately recapitulate TaG-EM barcode expression.

These are not independent sequencing libraries, but rather data from barcodes that were multiplexed in a single flow sort, 10x droplet capture, and sequencing library. Thus, there must be some other variable that explains the differential recovery of these barcodes.

• Fig 4A should appear earlier in the paper.

We have moved Figure 4A from the previous manuscript (a schematic showing the detailed design of the TaG-EM construct) to Figure 1A in the revised version.

**Reviewer #2 (Recommendations For The Authors):**
Minor:(1) There is a typo for Fig S13 figure legends: BC1, BC1, BC3... should be BC1, BC2, BC3.

Fixed.

**Reviewer #3 (Recommendations For The Authors):**
Comments to authors:(1) It would be great if the authors could provide an additional explanation on how these 29 barcode sequences were determined.

Response: This information is in the Methods section. For the original cloned plasmids:

“Expected construct size was verified by diagnostic digest with _Eco_RI and _Apa_LI. DNA concentration was determined using a Quant-iT PicoGreen dsDNA assay (Thermo Fisher Scientific) and the randomer barcode for each of the constructs was determined by Sanger sequencing using the following primers:

SV40_post_R: GCCAGATCGATCCAGACATGA

SV40_5F: CTCCCCCTGAACCTGAAACA”

For transgenic flies, after DNA extraction and PCR enrichment (details also in the Methods section):

“The barcode sequence for each of the independent transgenic lines was determined by Sanger sequencing using the SV40_5F and SV40_PostR primers.”

(2) Why did the authors choose myr-GFP as the backbone instead of nls-GFP if the downstream application is to perform sequencing?

We initially chose myr::GFP as we planned to conduct single cell and not single nucleus sequencing and myr::GFP has the advantage of labeling cell membranes which could facilitate the characterization or confirmation of cell type-specific expression, particularly in the nervous system. However, we have considered making a version of the TaG-EM construct with a nuclear targeted GFP (thereby enabling “NucEM”). In the Discussion, we mention this possibility as well as the possibility of using a second nuclear-GFP construct in conjunction with TaG-EM lines is nuclear enrichment is desired:

“In addition, while the original TaG-EM lines were made using a membrane-localized myr::GFP construct, variants that express GFP in other cell compartments such as the cytoplasm or nucleus could be constructed to enable increased expression levels or purification of nuclei. Nuclear labeling could also be achieved by co-expressing a nuclear GFP construct with existing TaG-EM lines in analogy to the use of hexameric GFP described above.”

Minor comments:(1) Line 193, Supplemental Figure 4 should be Supplemental Figure 5

Fixed.

(2) Scale bars should be added in Figure 4, Supplemental Figures 6, 7, and 8A.

We have added scale bars to these figures and also included scale bars in additional Supplemental Figures detailing characterization of the gut driver lines.

(3) Were Figure 4C and Supplemental Figure 7 data stained with a GFP antibody?

No, this is endogenous GFP signal. This is now noted in the Figure legends.

(4) Line 220, specify the three barcode lines (lines #7, 8, 9) in the text.

Added this information.

Same for Lines 251-254. Line 258, which 8 barcode Gal4 line combinations?

(5) Line 994, typo: (BC1, BC1, BC3, and BC7)-> (BC1, BC2, BC3, and BC7)

Fixed.

(6) Figure 5 F, J and N, add EC-Gal4, EB-Gal4, and EE-Gal4 above each panel to improve readability.

We have added labels of the cell type being targeted (leftmost panels), the barcode, and the marker gene name to Figure 6 C-N.